# Spatiotemporal profiling of modification-specific proteome secretion uncovers an itaconation-activated tyrosine kinase

Wenjie Lu[1,2,3,4,5,8], Yanling Zhang[1,2,3,4,5,8], Xinrui Ni[6,7], Pian Wang[6,7], Shentian Zhuang ✉[6,7] & Wei Qin ✉[1,2,3,4,5]

Macrophages secrete diverse signaling proteins critical for intercellular communication and immune responses, processes tightly regulated by post-translational modifications (PTMs). Itaconate, an immunoregulatory metabolite produced in macrophages, induces widespread intracellular protein modification (itaconation), affecting pathways like the KEAP1-NRF2 axis and glycolysis. However, the impact of itaconation on the extracellular proteome and signaling remains poorly characterized. Herein, we introduce PTM-based secretome profiling (PBSP), a novel approach to identify secreted proteins bearing specific PTMs. The method employs a bioorthogonal probe to label modified proteins in live cells, followed by enrichment of labeled proteins from the culture medium upon secretion. We established a streamlined chemoproteomic workflow integrating spintip-based affinity purification (FISAP) with data-independent acquisition (DIA) mass spectrometry for enhanced sensitivity and coverage. This identified 818 macrophage-secreted itaconated proteins, among which 447 are exosome-dependent. Further biochemical analysis revealed that itaconation of Cys239 on FYN (a tyrosine kinase) enhances its kinase activity in macrophages. We finally demonstrate PBSP's versatility by profiling secreted proteins with other PTMs, including fumarate-induced succination. PBSP provides a powerful platform to explore PTM roles in protein secretion, offering insights into PTMs' regulatory functions in cell-cell communication.

Nearly half of the human proteome resides in the extracellular space. Proteins containing a secretion signal peptide are typically transported via the canonical secretory pathway through the endoplasmic reticulum (ER) and Golgi apparatus[1]. However, recent studies have revealed that a surprisingly large number of proteins lacking a signal peptide are also secreted through unconventional pathways, such as exosomes and migrasomes[2–4]. When activated during pathogen infection or tumor development, macrophages secrete a variety of signaling proteins, including cytokines, into the extracellular space through both conventional and unconventional pathways[5,6]. These secreted proteins can travel to different organs and interact with various cell types, triggering downstream signaling cascades in recipient cells[7]. For

[1]The State Key Laboratory of Membrane Biology, Tsinghua University, Beijing, China. [2]School of Pharmaceutical Sciences, Tsinghua University, Beijing, China. [3]Tsinghua-Peking Center for Life Sciences, Tsinghua University, Beijing, China. [4]MOE Key Laboratory of Bioorganic Phosphorus Chemistry & Chemical Biology, Tsinghua University, Beijing, China. [5]Beijing Frontier Research Center for Biological Structure, Tsinghua University, Beijing, China. [6]Institute of Translational Medicine, China Pharmaceutical University, Nanjing, China. [7]Center for Infectious Medicine and Vaccine Research, School of Basic Medicine and Clinical Pharmacy, China Pharmaceutical University, Nanjing, China. [8]These authors contributed equally: Wenjie Lu, Yanling Zhang. ✉e-mail: shentianzhuang@cpu.edu.cn; weiqin@tsinghua.edu.cn

example, extracellular vesicles (EVs) released by activated macrophages can deliver CDC42, a plasma membrane-associated GTPase, to recipient cells, thereby triggering intracellular immune responses[8].

Simultaneously, macrophages produce significant amounts of itaconate in the mitochondria, reaching millimolar concentrations in pathogen-infected macrophages[9,10]. Itaconate has also been shown to regulate immune responses at the host-pathogen interface and within the tumor microenvironment, partly through its ability to modify cysteine residues on proteins[11,12]. Key immune regulators, such as KEAP1[13], ALDOA[14], JAK1[15], and GSDMD[16,17], are among the substrates of itaconation. These modifications influence critical pathways, including the NRF2 anti-inflammatory and antioxidant signaling pathway, glycolysis, and pyroptosis. However, current chemoproteomic and functional studies on itaconation have primarily focused on intracellular proteins[8–17]. The potential modifications and regulatory roles of itaconate in secreted proteins, which make up nearly half of the proteome and play essential roles in macrophage biology and tumor microenvironment, remain largely unexplored.

We previously developed chemoproteomic strategies to systematically map itaconate modifications, identifying thousands of itaconated proteins in macrophages[14,18]. However, these studies have not yet elucidated the spatiotemporal dynamics of itaconate-modified proteins. Recent advancements in spatiotemporally resolved proteomic technologies, particularly proximity labeling-based approaches such as TransitID, now enable unbiased mapping of protein trafficking and secretion[19–23]. These methods employ a "pulse-chase" strategy to label intracellular proteins in specific cell types, followed by tracking and collecting labeled proteins in the extracellular space. Despite their utility, proximity labeling technologies currently lack sufficient resolution to investigate the secretion of specific proteoforms, such as proteins modified by distinct post-translational modifications (PTMs). While PTMs like phosphorylation, glycosylation, and acylation are known to regulate protein trafficking and secretion[24–26], existing studies predominantly focus on individual proteins or pathways. Consequently, there is an urgent need for methods capable of large-scale, unbiased analysis of PTM-regulated protein secretion and communication events.

We introduce PBSP, a spatiotemporally resolved chemoproteomic workflow designed to map secreted proteins harboring specific PTMs. The method employs chemical probes to pulse-label intracellular proteins with a defined PTM, followed by probe washout and subsequent collection of labeled proteins from the culture medium after a customizable chase period (Fig. 1a). This temporal control ensures that detected proteins are modified intracellularly prior to

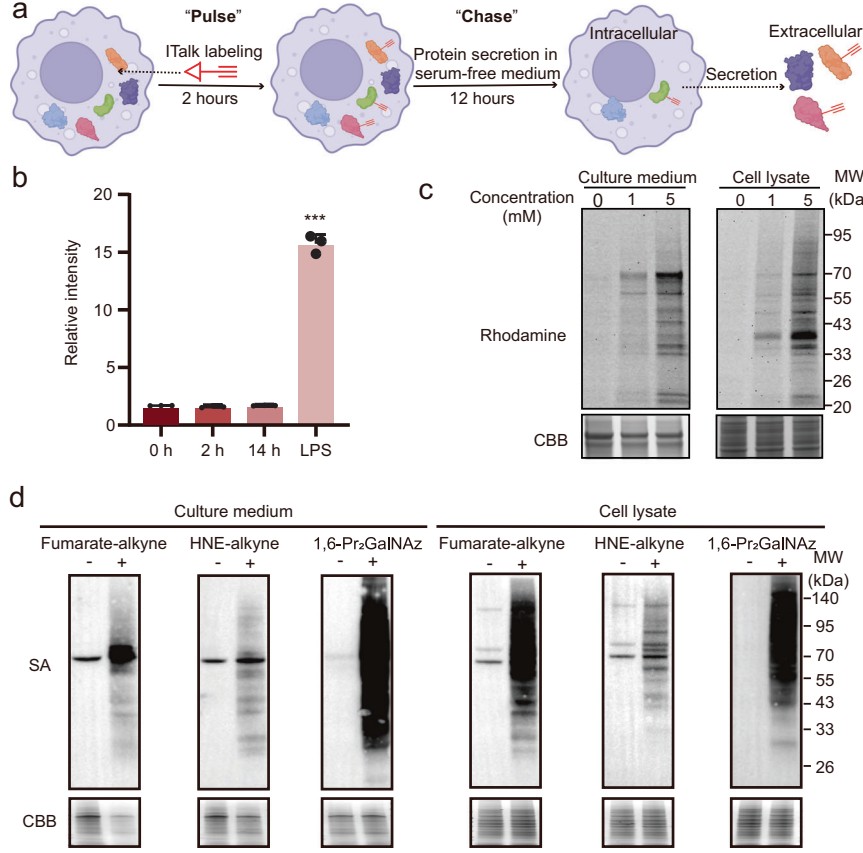

**Fig. 1 | Development of PBSP to track the spatiotemporal secretion of itaconated proteins in living macrophages. a** Schematic illustrating the pulse-chase labeling of itaconated secretomes with ITalk. Macrophages were pulse-labeled with ITalk for 2 hours, followed by probe washout and a 12 hours chase period in serum-free medium. Secreted proteins in the medium were then collected for click reaction-based detection. Created in BioRender. Chu, l. (2025) https://BioRender.com/p66bt7z. **b** Effect of the PBSP workflow on itaconate levels. Metabolites were extracted from Raw264.7 cells after the complete pulse-chase procedure and itaconate was quantified by LC-MS/MS. For the positive control, cells were stimulated with LPS (100 ng/mL) for 14 hours. Quantification was performed from three biological replicates and the error bars show mean ± SD. ***$p < 0.001$ (two-sided Student's t-test). $p$-values: 0.11317 (2 hours), 0.41508 (14 hours), 0.00001 (Lps 12 hours). **c** In-gel fluorescence detection of ITalk labeling in the culture medium (left) or cell lysates (right). Raw264.7 cells were treated with 0, 1, or 5 mM ITalk for 2 hours, followed by a 12 hours chase in serum-free medium. **d** Streptavidin blot (SA-blot) detection of distinct probe labeling in the culture medium (left) or cell lysates (right). For (**c**, **d**), both the culture medium and cells were subjected to protein extraction and click reaction with azide-Rhodamine or azide-biotin. Labeling intensity was determined by in-gel fluorescence scanning or SA-blot, and Coomassie Brilliant Blue (CBB) staining was used to confirm equal protein loading. Three replicates were performed with similar results. Corresponding uncropped images are shown in the Source Data file.

secretion through conventional or unconventional pathways. By integrating spintip-based affinity purification (FISAP) with data-independent acquisition (DIA) mass spectrometry, we developed a high-sensitivity workflow that identified 818 macrophage-secreted proteins modified by itaconation, including a subset trafficked via exosome-dependent secretion. Strikingly, we uncovered itaconation of the tyrosine kinase FYN at Cys239 in macrophages, with subsequent exosome-mediated secretion. This modification enhances FYN activity, as indicated by reduced inhibitory phosphorylation at Tyr531. Furthermore, we demonstrate the versatility of PBSP by adapting it to profile the secretion dynamics of proteins modified by alternative PTMs, such as succination.

## Results

### Establishment of PBSP

To establish PBSP, we utilized itaconation as a proof-of-concept system, leveraging the bioorthogonal probe itaconate-alkyne (ITalk) to label itaconated proteins in live macrophages[14]. Following secretion of ITalk-labeled proteins, the culture medium was subjected to click chemistry with azide-biotin, enabling enrichment of labeled secreted proteins and subsequent proteomic analysis (Supplementary Fig. 1a).

First, we conducted ITalk labeling in living Raw264.7 cells, and in-gel fluorescence scanning after the click reaction with azide-Rhodamine revealed both concentration- and time-dependent ITalk labeling (Supplementary Fig. 1b). Next, we assessed the pulse-chase labeling of ITalk. To achieve high temporal resolution, Raw264.7 cells were pulse-labeled with 1 or 5 mM ITalk for a short period (2 hours). The probe-containing medium was then removed, and the cells were washed extensively with PBS. Subsequently, cells were subjected to a defined chase period (e.g., 12 hours) in serum-free medium lacking ITalk, following an established protocol for secretome analysis[22]. We used serum-free medium to prevent interference from the high abundance of serum proteins during subsequent enrichment steps. Following the entire pulse-chase procedure, metabolites were extracted from the cells for the quantification of itaconate by liquid chromatography-tandem mass spectrometry (LC-MS/MS) (Fig. 1b). The results confirmed that the labeling process does not interfere with itaconate metabolism. After the chase period, both the cells and the culture medium were collected for click chemistry-based in-gel fluorescence detection. We observed concentration-dependent ITalk labeling in the macrophage secretome, with labeling patterns that were distinctly different from those observed in the cells (Fig. 1c). These results suggest that ITalk labeling is compatible with the pulse-chase format and enables the detection of itaconated proteins in the secretome.

To evaluate the versatility of PBSP, we conducted metabolic/chemical labelling of glycosylation, carbonylation, and succination using 1,6-di-O-propionylated-N-azidoacetylgalactosamine (1,6-Pr$_2$GalNAz)[27,28], 4-hydroxy Nonenal-alkyne (HNE-alkyne)[29], and fumarate-alkyne[30], respectively (Supplementary Fig. 1c). Raw264.7 cells were subjected to probe pulse labelling followed by a chase in serum-free medium. In-gel fluorescence scanning of secreted proteins extracted from the medium revealed distinct labelling patterns for each probe (Fig. 1d), indicating a diverse population of proteoforms bearing different PTMs. Interestingly, GalNAz most extensively labels secreted proteins, consistent with the biological knowledge that many secreted proteins undergo mucin-type O-linked glycosylation[31,32]. This experiment highlights PBSP's potential to study protein secretion mediated by distinct PTMs.

### Development of a streamlined proteomic workflow for PBSP

We next aimed to establish an optimized proteomic workflow to identify itaconated proteins in the secretome. After concentrating the culture medium using a 10 kDa filter, the secreted proteins were resuspended in RIPA buffer and subjected to a click reaction with azide-biotin, allowing biotinylation of the ITalk-labeled proteins. Since only a small amount of proteins (~100 μg) were secreted into the culture medium during the 12 hours chase period, we needed to modify our standard sample processing workflow to minimize protein loss and ensure compatibility with low-input samples. To address this, we utilized the FISAP platform[33], which incorporates a custom C18 tip loaded with streptavidin beads (Fig. 2a). The biotinylated proteins bind to the streptavidin beads on the tip, while unmodified proteins are removed by centrifugation. On-bead trypsin digestion was then performed, and the resulting peptides were released into the C18 phase. The peptides were desalted directly on the tip before being eluted for LC-MS/MS analysis.

To further enhance the sensitivity and reproducibility of PBSP, we employed DIA mode, as opposed to data-dependent acquisition (DDA), the method used in previous itaconation mapping studies[14,18] (Fig. 2b). DDA is prone to random data loss due to its semi-stochastic sampling nature, making it challenging to compare data across independent MS experiments. In contrast, DIA offers a more robust approach by simultaneously co-fragmenting all co-eluting peptide ions within predefined mass-to-charge (m/z) windows, enabling the concurrent measurement of their corresponding fragments[34]. This method provides more comprehensive coverage of precursor peptide ions, irrespective of their intensities, and ensures more accurate quantitation with enhanced reproducibility.

We then applied the PBSP workflow to the ITalk-labeled proteins from the culture medium, along with a negative control that omitted ITalk treatment (Supplementary Fig. 2a). The DIA data were analyzed using DIA-NN[35], a "library-free" approach that does not require spectral libraries generated by DDA methods. The enrichment ratios across replicates were also highly correlated, indicating strong quantification accuracy (Fig. 2c). We defined proteins with an enrichment ratio (ITalk vs. N.C.) greater than 2 as enriched, retaining those whose $p$-value is less than 0.05 in three biological replicates (Fig. 2d and Supplementary Fig. 2b). This resulted in the identification of 818 ITalk-labeled secreted proteins (Supplementary Table 1).

### Analysis of ITalk-labeled secreted proteins

We found that 494 out of 818 itaconated secreted proteins had previously been identified as itaconated by S-glycosylation-based competitive profiling[14] or ITalk-based direct mapping[18], indicating that the itaconated secreted proteins identified by PBSP are highly specific to itaconation (Fig. 3a). Several functional itaconated substrates, such as Gasdermin-D (GSDMD), Kelch-like ECH-associated protein 1 (KEAP1), Eukaryotic translation initiation factor 4 gamma 1 (EIF4G1), Fructose-bisphosphate aldolase A (ALDOA), L-lactate dehydrogenase A chain (LDHA) and Glutaredoxin-1 (GLRX1), were also identified as secreted with itaconation (Supplementary Fig. 3a). Moreover, the majority of the itaconated secreted proteins (70%) are annotated as secretory proteins[36], suggesting that PBSP also has high specificity for protein secretion (Fig. 3b). KEGG pathway analysis of these itaconated proteins revealed enrichment in pathways related to the proteasome, carbon metabolism, and the biosynthesis of amino acids (Fig. 3c). In comparison, newly identified itaconated proteins showed significant enrichment in pathways such as the lysosome, ribosome, glycan degradation and COVID-19 (Fig. 3d). Notably, low itaconate levels have been shown to correlate with COVID-19 disease severity, and itaconate derivatives exhibit significant antiviral effects[37,38]. We identified many critical proteins involved in these pathways that are both itaconated and secreted, including RNA sensor antiviral innate Immune response receptor RIG-I (RIG-I), Mitogen-activated protein kinase 1 (MAPK1), and MAP kinase-activated protein kinase 3 (MAPK3) (Fig. 3e). Lysosomes, which play a central role in macrophage-mediated bacterial killing and intercellular communication, have also been functionally linked with itaconate[39]. We found various lysosomal enzymes, such as ATPase, phosphatase, and cathepsin, to be itaconated secreted proteins

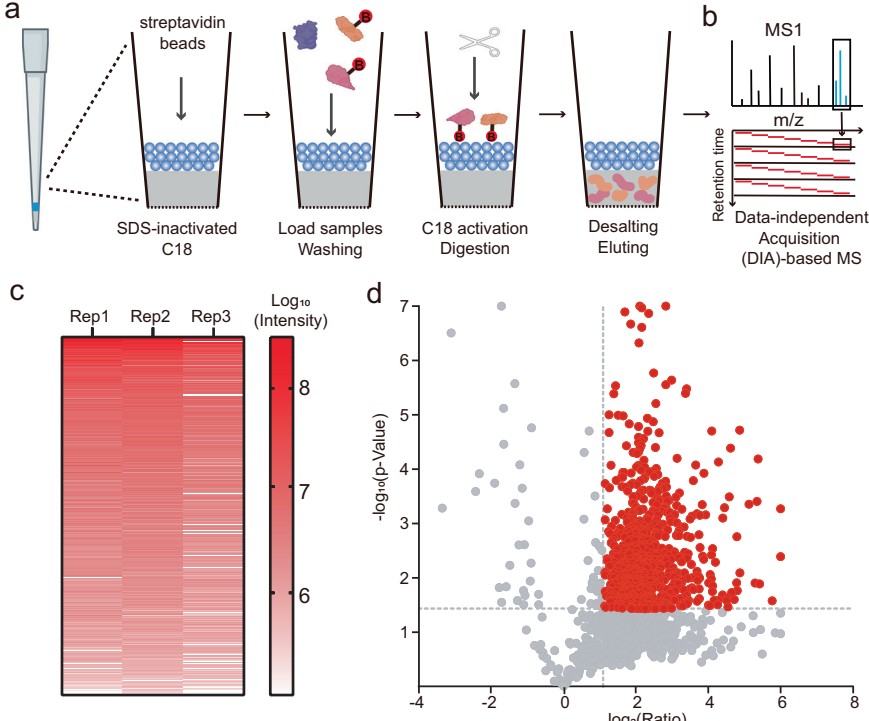

**Fig. 2 | Establishment of a streamlined chemoproteomic workflow for PBSP profiling. a** Schematic of the FISAP platform. A custom C18 tip, pre-loaded with streptavidin beads, was first treated with sodium dodecyl sulfate (SDS) to inactivate the C18. ITalk-labeled proteins extracted from the culture medium were click-reacted with azide-biotin and then incubated with the streptavidin beads for enrichment. After removing unlabeled proteins by centrifugation, the C18 was re-activated, and on-bead trypsin digestion was performed. The released peptides were desalted directly by the C18, followed by elution for DIA-based LC-MS/MS analysis. Created in BioRender. Chu, l. (2025) https://BioRender.com/e55b8pm. **b** Schematic of the DIA-based LC-MS/MS analysis. **c** Correlation of protein intensities quantified by the PBSP workflow across three biological replicates. **d** Volcano plot showing the comparative enrichment of proteins between ITalk labeling group and control group by PBSP. Itaconated secreted proteins are highlighted in red. *p*-values were determined by two-sided Student t-tests.

(Fig. 3f). Additionally, key proteins involved in ubiquitination process were secreted with itaconation, such as Ubiquitin carboxyl-terminal hydrolase 5 (USP5), USP14, USP19, USP39 and Probable ubiquitin carboxyl-terminal hydrolase FAF-X (USP9X), potentially mediating signaling between itaconate and protein degradation (Fig. 3g). These analyses reveal that the itaconated secreted proteins identified by PBSP provide new insights into the roles of itaconate across various aspects of the immune system.

To evaluate the broad applicability of our approach, we performed PBSP profiling in immortalized bone marrow-derived macrophages (iBMDMs) under the same ITalk labeling conditions. Using identical filtering criteria, we identified 54 itaconated secreted proteins (Supplementary Fig. 3b, Supplementary Table 2). Among these, 38 are known secreted proteins (70.4%) and 37 were previously reported as itaconated (68.5%), mirroring the specificity we observed in Raw264.7 cells (Supplementary Fig. 3c, d). These results demonstrate that PBSP is broadly applicable and can reveal cell type-specific heterogeneities.

**Identification of exosome-dependent ITalk-labeled secreted proteins**

Exosomes are membrane-enclosed vesicles naturally released by nearly all cell types, including macrophages, and serve as an important mechanism for unconventional protein secretion[40]. These vesicles are recognized as crucial carriers of biological information, transferring their contents - such as proteins - from parent cells to recipient cells[41]. For example, exosomes derived from skeletal muscle have been shown to suppress macrophage inflammatory responses by upregulating the immune-responsive gene 1 (IRG1) pathway, which boosts itaconate production[42]. The functions of macrophage-derived exosomes in various disease contexts have also been extensively studied, with growing evidence suggesting that these exosomes play a critical role in disease progression[43]. However, it remains largely unknown whether itaconated proteins in macrophages can be incorporated into exosomes, and how itaconation might influence the function of these proteins in intercellular communication.

Inspired by the use of TransitID to investigate exosome-mediated protein transfer between macrophages and cancer cells by employing the exosome inhibitor GW4869[44], we aimed to apply PBSP to identify itaconated proteins secreted via exosomes. Raw264.7 cells were treated with ITalk and GW4869 for 2 hours, followed by a 12 hours chase in serum-free medium containing GW4869 (Fig. 4a). Exosome-inhibited secreted proteins that were itaconated were then enriched and analyzed using our streamlined PBSP workflow, which demonstrated excellent consistency across biological replicates (Supplementary Fig. 4a). We conducted further quantitative comparisons between the basal and GW4869-treated conditions, identifying 447 itaconated proteins whose secretion significantly decreased upon exosome inhibition (Fig. 4b and Supplementary Table 3). The majority of these exosome-mediated, itaconated secreted proteins are known cargoes of exosomes[45], highlighting the specificity of this exosome-dependent dataset (Fig. 4c). Among the identified proteins, a functionally significant itaconated protein, GSDMD, was also found to be secreted via exosomes, suggesting that itaconation may further influence its functional activity in recipient cells (Fig. 4d). In comparison, LDHA exhibits unchanged protein secretion upon exosome inhibition, suggesting it undergoes exosome-independent secretion.

Gene Ontology (GO) analysis of their biological processes revealed the enrichment of terms such as protein transport, protein folding, and protein stabilization (Fig. 4e). Four components of the COP9 signalosome complex (CSN)—a deNEDDylase that regulates the

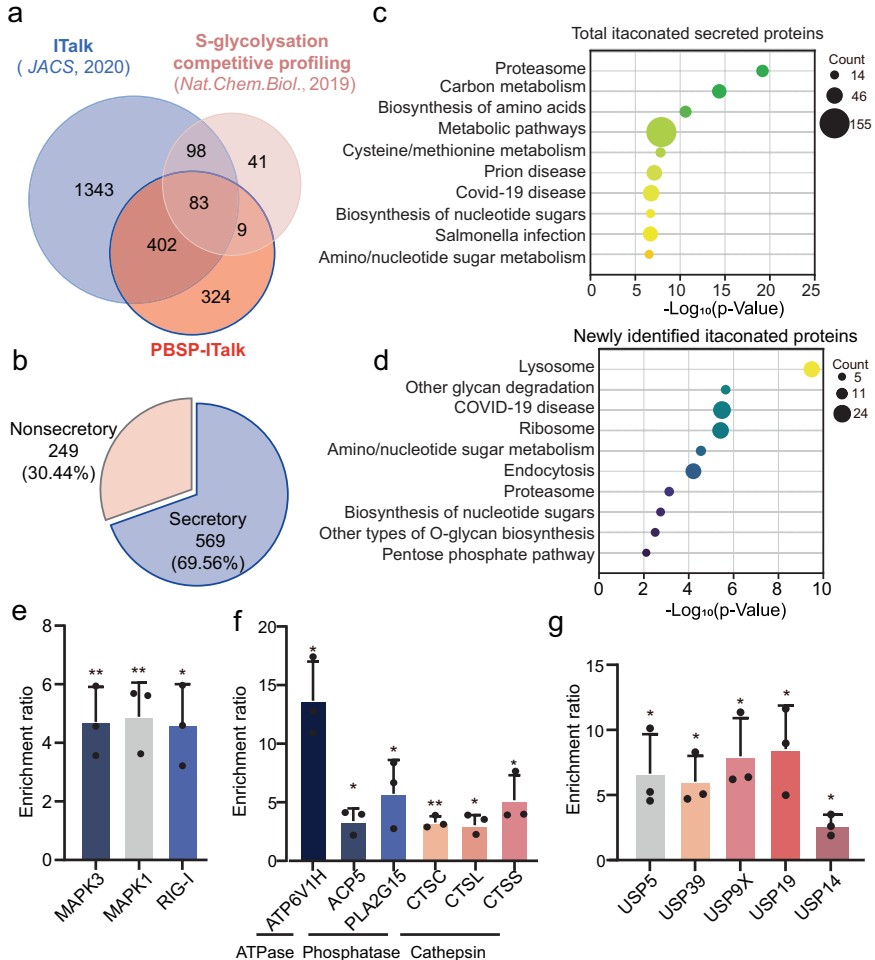

**Fig. 3 | Analysis of itaconated secreted proteins identified by PBSP.**
**a** Comparison of itaconated proteins identified by PBSP with those previously identified through intracellular ITalk labeling and S-glycosylation-based competitive cysteine profiling. **b** Proportion of known secretory proteins among the itaconated secreted proteins identified by PBSP. **c** KEGG pathway analysis of the itaconated secreted proteins identified by PBSP with one-sided $p$-values from Fisher's Exact test. **d** KEGG pathway analysis of the newly identified itaconated proteins with one-sided $p$-values from Fisher's Exact test. **e** Enrichment ratios of representative itaconated secreted proteins involved in COVID-19, including the RNA sensor RIG-I, MAPK1, and MAPK3. $p$-values: 0.00571 (MAPK3), 0.00448 (MAPK1), 0.01017 (RIG-I). **f** Enrichment ratios of representative itaconated secreted enzymes in the lysosome. $p$-values: 0.00277 (ATP6V1H), 0.01711 (ACP5), 0.04596 (PLA2G15), 0.00152 (CTSC), 0.01452 (CTSL), 0.02720 (CTSS). **g** Enrichment ratios of representative itaconated secreted proteins involved in ubiquitination pathway. $p$-values: 0.03241 (USP5), 0.01159 (USP39), 0.01457 (USP9X), 0.01744 (USP19), 0.02384 (USP14). In (**e**–**g**), the error bars show mean ± SD from three biological replicates. *$p < 0.05$, **$p < 0.01$ (two-sided Student's t-test).

ubiquitination activity of cullin-RING E3 ligases and modulates various targets in inflammation[46]—were identified as exosome-mediated, itaconated secreted proteins (Fig. 4f). These components may be itaconated and secreted in an exosome-dependent manner, likely influenced by their complex interactions. Additionally, several key proteins involved in apoptosis, including cyclin-dependent kinase 1 (CDK1), were also identified as exosome-mediated, itaconated secreted proteins (Fig. 4g). These proteins could play a role in regulating the apoptotic processes in recipient cells, with their itaconation potentially contributing to these effects. Among the exosome-mediated itaconated secreted proteins, we identified 204 novel itaconated substrates (Supplementary Fig. 4b). These newly identified itaconated proteins show strong associations with lysosomal function and amino/nucleotide sugar metabolism (Fig. 4h), highlighting potential itaconation-regulated biological pathways not captured by whole-cell profiling.

**Itaconation of FYN at Cys239 promotes FYN kinase activity**
From novel itaconated proteins, we selected two tyrosine-protein kinases—Tyrosine-protein kinase Fyn (FYN) and vascular endothelial

growth factor receptor 1 (FLT1)—for further biochemical validation[47–50]. ITalk labeling in Raw264.7 cells, followed by enrichment and Western blot detection, confirmed the itaconation of both proteins (Fig. 5a). We also purified His-tagged full-length recombinant FYN from *E. coli* (Supplementary Fig. 5a). Incubation of recombinant FYN with ITalk revealed concentration-dependent labeling (Supplementary Fig. 5b). Notably, ITalk labeling was efficiently inhibited by pretreatment with iodoacetamide, a thiol-reactive cysteine-blocking reagent, confirming cysteine-dependent modification (Fig. 5b). To identify the specific cysteine residues in FYN that are modified by itaconate, we treated recombinant FYN with itaconate, digested it with trypsin, and analyzed the peptides by LC-MS/MS. This analysis identified six cysteine residues—Cys239, Cys240, Cys246, Cys404, Cys487, and Cys491—as potential sites of itaconation (Supplementary Fig. 5c). However, due to their close proximity, the MS/MS spectra could not unambiguously distinguish whether itaconation occurred on Cys239 or Cys240 (Fig. 5c).

FYN contains nine cysteine residues in total. Among these, the N-terminal Cys3 and Cys6 residues are palmitoylated, facilitating membrane anchoring[51]. To confirm FYN cysteines susceptible to

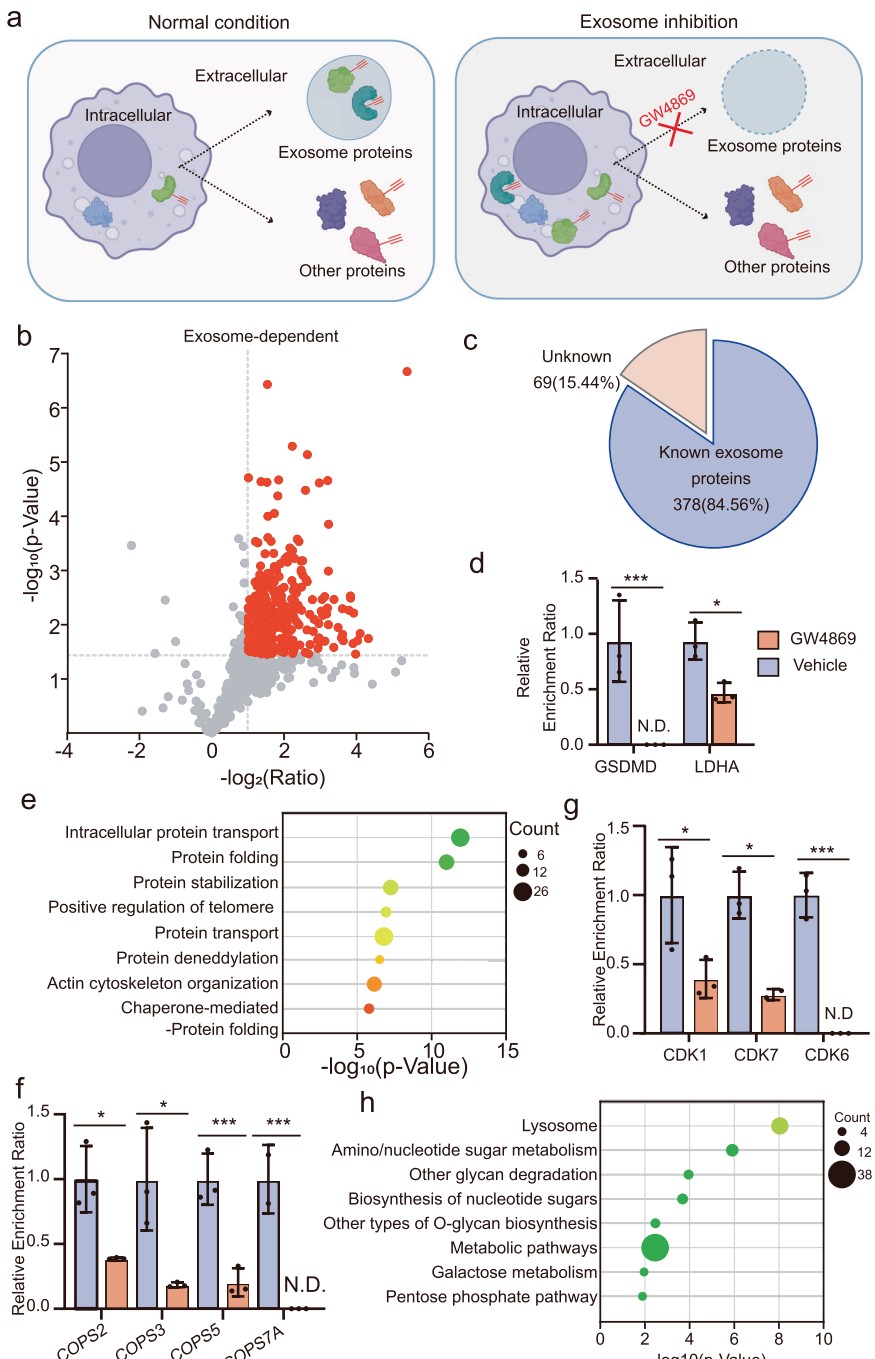

**Fig. 4 | Identification of exosome-dependent itaconated secreted proteins by PBSP. a** Schematic of PBSP profiling under normal and exosome inhibition conditions. The exosome inhibitor GW4869 was used to block exosome secretion, reducing the enrichment of exosome-dependent itaconated secreted proteins. Created in BioRender. Chu, l. (2025) https://BioRender.com/wudkzfu. **b** Volcano plot showing the comparative enrichment of proteins between normal and exosome inhibition conditions. Exosome-dependent itaconated secreted proteins are highlighted in red. *p*-values were determined by two-sided Student t-tests. **c** Proportion of known exosome proteins within the exosome-dependent itaconated secreted proteins. **d** Enrichment ratios of exosome-dependent itaconated proteins (GSDMD) and exosome-independent itaconated proteins (LDHA) under normal and exosome inhibition conditions. *p*-values: 0 (GSDMD), 0.01300 (LDHA).

**e** Gene Ontology (GO) biological process analysis of the exosome-dependent itaconated secreted proteins identified by PBSP with one-sided *p*-values from Fisher's Exact test. **f** Enrichment ratios of four components of the COP9 signalosome complex (CSN). *p*-values: 0.01396 (COPS2), 0.02366 (COPS3), 0.00364 (COPS3), 0 (COPS7A). **g** Enrichment ratios of representative proteins involved in apoptosis, including CDK1, CDK6, and CDK7, under normal and exosome inhibition conditions. *p*-values: 0.04851 (CDK1), 0.01117 (CDK7), 0 (CDK6). N.D., not determined. In (**d**, **f**, **g**), the error bars show mean ± SD from three biological replicates. *\*p* < 0.05, *\*\*\*p* < 0.001 (two sided Student's t-test). **h** KEGG analysis of 204 novel itaconated proteins in the exosome-dependent itaconated secreted proteins with one-sided *p*-values from Fisher's Exact test.

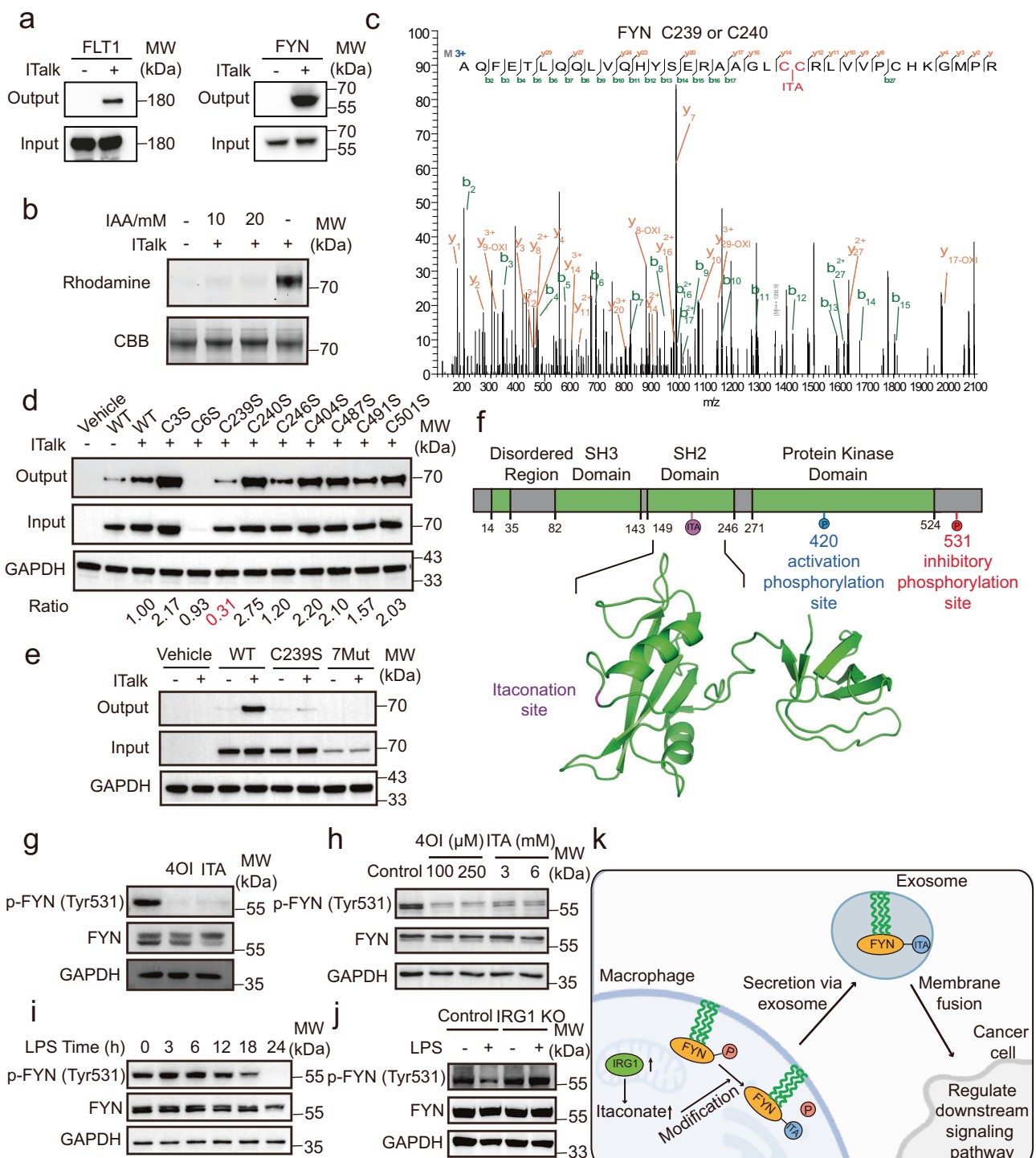

**Fig. 5 | Itaconation at Cys239 enhances FYN kinase activity. a** Western blotting validation of novel itaconated proteins (FYN, FLT1) identified by PBSP. **b** Inhibition of ITalk labeling on purified FYN by iodoacetamide (IAA) pre-treatment. Purified FYN (20 μM) was incubated with 10 mM or 20 mM iodoacetamide for 30 min, then treated with 1 mM ITalk for 1 hour. **c** Tandem MS spectra of peptides containing itaconation potentially on Cys239 of FYN. **d** ITalk labeling in FYN single-cysteine mutants. **e** ITalk labeling in a FYN mutant lacking all seven cysteines. In (**d**, **e**), HEK293T cells expressing each FYN mutant were treated with 1 mM ITalk for 6 hours. Lysates were subjected to click reaction with azide-biotin, streptavidin enrichment, and anti-MYC blotting. **f** Domain architecture and structural location

of the FYN itaconation site (Cys239, SH2 domain; PDB: 3UF4). **g** Reduced FYN Tyr531 phosphorylation following 24 hours treatment with 3 mM itaconate or 500 μM 4-OI in Raw264.7 cells. **h** The impact of itaconate and 4-OI on FYN phosphorylation at Tyr531 in BMDMs. **i** Time-dependent reduction of FYN Tyr531 phosphorylation in Raw264.7 cells treated with 100 ng/mL LPS. **j** FYN Tyr531 phosphorylation in WT and IRG1 KO cells following LPS treatment. **k** Proposed hypothesis on how itaconation of FYN regulates intercellular communications. Created in BioRender. Chu, l. (2025) https://BioRender.com/f0mqqeh. In (**a**, **b**, **d**, **e**, **g**, **h**, **i**, **j**), three replicates were performed with similar results. Corresponding uncropped images are shown in the Source Data file.

itaconate modification in a cellular context, we individually mutated each of the nine cysteines and transfected the mutants into HEK293T cells. Following treatment with ITalk, cell lysates were click-labeled with azide-biotin, enriched via streptavidin pulldown, and analyzed by anti-MYC Western blotting. Strikingly, only mutation of Cys239 substantially reduced ITalk labeling, identifying it as FYN's primary itaconation site (Fig. 5d). Mutation of Cys6 abolished FYN expression entirely, likely due to disrupted palmitoylation and subsequent loss of membrane anchoring. Finally, we created a mutant retaining only the two palmitoylation sites (Cys3 and Cys6) while all other cysteines were mutated. This mutant eliminated ITalk labeling completely, although the protein's stability was severely impaired (Fig. 5e).

FYN is a non-receptor cytoplasmic tyrosine kinase belonging to the Src family kinases and plays a role in multiple signal transduction pathways[52]. Structurally, it contains an SH3 domain, an SH2 domain, and a C-terminal protein kinase domain[53]. FYN activity is regulated by dynamic transitions between active and inactive conformations, governed by phosphorylation and dephosphorylation of two critical tyrosine residues: Tyr420 (activation site) and Tyr531 (inhibitory site)[54]. In its active "open" conformation, Tyr420 is phosphorylated while Tyr531 remains unphosphorylated, enabling the catalytic domain to engage with substrates. Conversely, phosphorylation of Tyr531 within the negative regulatory tail induces an inactive "closed" conformation, where the tail binds the SH3/SH2 domains, blocking substrate access. Notably, the itaconation site on FYN is positioned at a highly exposed region of the SH2 domain that might interfere with SH2-tail interactions (Fig. 5f), prompting us to investigate whether itaconation influences adjacent phosphorylation events and modulates FYN activity.

To test this, Raw264.7 cells were treated with 3 mM itaconate or 500 µM 4-octyl itaconate (4-OI) for 24 hours. Both treatments significantly reduced FYN Tyr531 phosphorylation (Fig. 5g), suggesting itaconate promotes FYN activation. This effect was also confirmed on mouse bone marrow-derived macrophages (BMDMs) (Fig. 5h). To assess endogenous regulation, LPS (lipopolysaccharide) stimulation induced itaconate production over time. Consistent with exogenous treatment, 24 hours LPS exposure abolished Tyr531 phosphorylation (Fig. 5i). Intriguingly, this effect was time-dependent, as Tyr531 phosphorylation remained unchanged at 18 hours despite LPS stimulation. Given that itaconate levels peak 24 hours post-LPS[55], these results imply that FYN activation requires a threshold itaconate concentration. To confirm that the suppression of FYN phosphorylation by LPS is mediated through itaconate (Supplementary Fig. 5d), we examined *IRG1*-knockout Raw264.7 cells. We found that LPS treatment did not reduce FYN phosphorylation at Tyr531 in the absence of IRG1 (Fig. 5j), indicating that endogenously produced itaconate via IRG1 is required for the LPS-mediated regulation of FYN. FYN phosphorylates diverse substrates to regulate downstream signaling, including nuclear factor erythroid 2-related factor 2 (NRF2)—a transcription factor critical for activating chemoprotective genes during electrophilic stress[56]. We propose that in multicellular systems like the tumor microenvironment, macrophages may release exosomes containing itaconated FYN, enabling this activated kinase to regulate downstream signaling in recipient cancer cells (Fig. 5k). Supporting this, we detected FYN in macrophage-derived extracellular vesicles (Supplementary Fig. 5e). Collectively, these results demonstrate that itaconation at Cys239 enhances FYN kinase activity and may regulate intercellular communication.

### Identification of succinated secreted proteins via PBSP

Fumarate, another immunoregulatory metabolite containing an α,β-unsaturated carboxylic acid group, non-enzymatically modifies cysteine residues to form protein succination—a mechanism analogous to itaconation[57]. Interestingly, itaconation and succination share

protein substrates, regulate overlapping signaling pathways, and are both implicated in immune responses and tumor microenvironment regulation[16,17,58]. Dimethyl fumarate, a clinically used immunomodulatory drug for multiple sclerosis and psoriasis, exerts its effects partly through protein succination and functional modulation[59]. Despite this, the identity of succinated secreted proteins and their role in intercellular communication remain largely unexplored. Leveraging the versatility of PBSP, we applied this platform to map succinated secreted proteins in macrophages.

We labeled succinated proteins using the alkynyl analogue fumarate-alkyne[30]. Raw264.7 cells were treated with 1 mM fumarate-alkyne for 4 hours, followed by a 12 hours chase in serum-free medium. Proteins labeled with fumarate-alkyne were collected from the medium and identified using FISAP coupled with DIA-based LC-MS/MS (Fig. 6a). Based on highly correlated enrichment ratios across replicates (Supplementary Fig. 6), we identified succinated secreted proteins as those with a significant enrichment (ratio >2, $p < 0.05$) over unlabeled controls (Fig. 6b, Supplementary Table 4). Of these, approximately 67% are annotated as secreted (Fig. 6c). KEGG pathway analysis revealed enrichment in complement and coagulation cascades, glutathione metabolism, and proteasome function pathways, which are functionally linked to fumarate (Fig. 6d). Notably, succinated secreted proteins showed minimal overlap with itaconated counterparts despite similar modification chemistries (Fig. 6e), suggesting PTM-specific regulatory mechanisms for substrate secretion.

Using PBSP with the exosome inhibitor GW4869, we identified 25 exosome-dependent succinated secreted proteins exhibiting significantly reduced enrichment upon inhibition (Fig. 6f, Supplementary Table 5). This group included known fumarate-related proteins[60] such as MMS19 (a nucleotide excision repair protein) and LIN7C (lin-7 homolog C), and 21 of the 25 proteins were known exosomal components (Fig. 6g), underscoring PBSP's specificity for pathway-dependent secretion. KEGG analysis highlighted enrichment in glutathione metabolism, drug metabolism, and chemical carcinogenesis (Fig. 6h). Collectively, these results demonstrate PBSP's versatility for profiling secretion events mediated by diverse PTMs.

## Discussion

Nearly half of the human proteome resides extracellularly, with numerous proteins dynamically shuttling between cells to orchestrate intercellular communication. As pivotal immune sentinels, macrophages respond to environmental stimuli (e.g., pathogens, cytokines) by secreting substantial quantities of signaling proteins[15]. While proximity labeling techniques like TransitID enable mapping of general protein trafficking and secretion, they lack resolution for specific proteoforms – particularly proteins bearing distinct PTMs. Our PBSP platform bridges this critical gap by enabling precise tracking of PTM-defined protein secretion. We demonstrate its utility across glycosylation, carbonylation, succination, and itaconation, with inherent adaptability to any PTM traceable via bioorthogonal chemical probes.

Cysteine itaconation represents an emerging immunoregulatory PTM that modifies key signaling nodes within macrophages, thereby modulating inflammatory pathways. Current research has predominantly focused on intracellular itaconation targets, leaving the landscape of secreted itaconated proteins largely unexplored. Applying PBSP, we unveil comprehensive insights into this secretory dimension. Our dataset not only confirms known itaconated proteins (KEAP1, GSDMD) are secreted in modified form – a previously unrecognized phenomenon – but also identifies novel substrates missed by intracellular studies[14,18], likely due to rapid export post-modification. Crucially, by integrating pathway-specific inhibitors (e.g., GW4869 for exosomes), we distinguished 447 exosome-dependent itaconated secreted proteins. Parallel PBSP analysis of succination revealed non-overlapping secretory

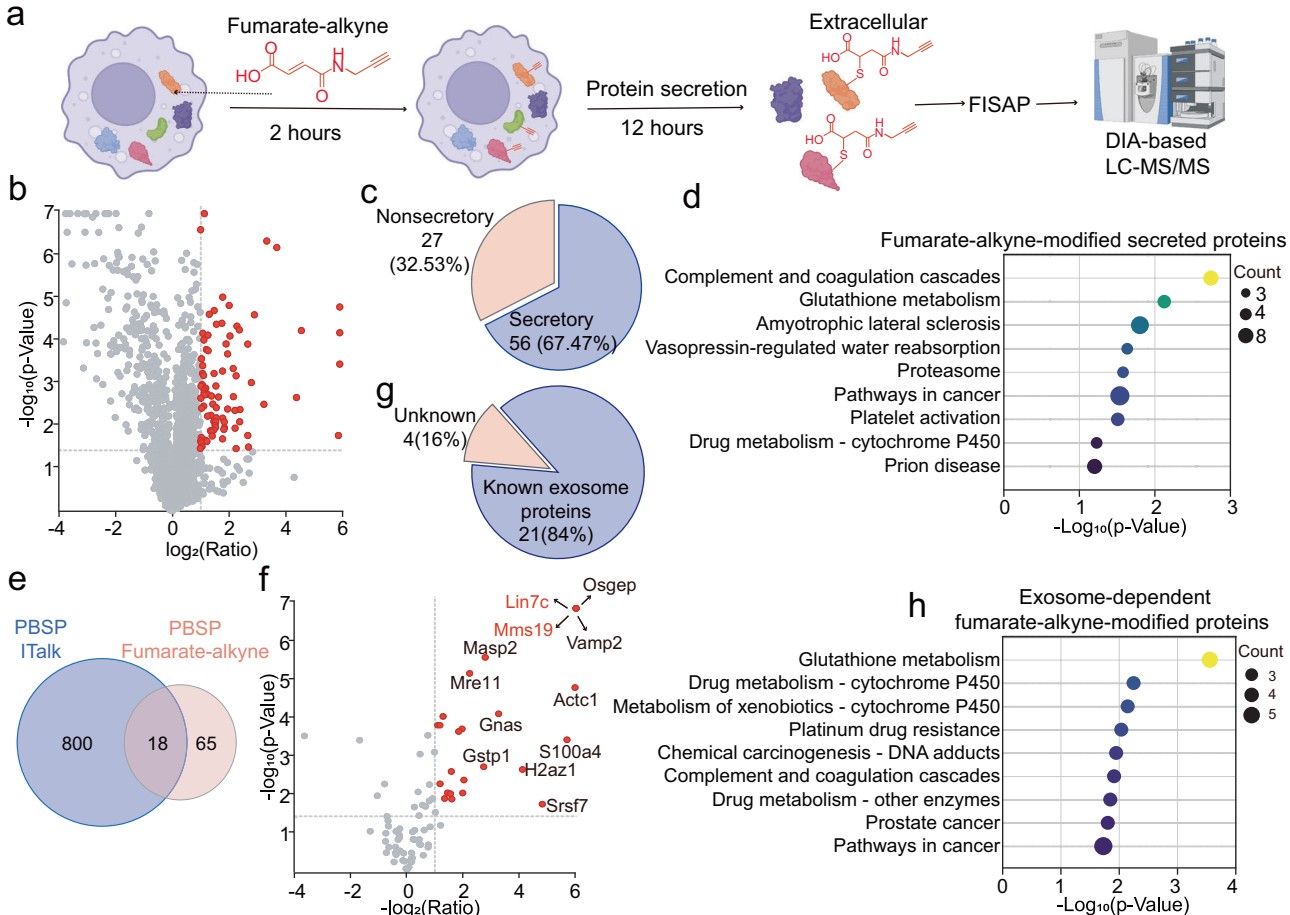

**Fig. 6 | Identification of succinated secreted proteins via PBSP. a** Workflow for mapping succinated secreted proteins with PBSP. Created in BioRender. Chu, l. (2025) https://BioRender.com/yx6pwe1. **b** Volcano plot showing the comparative enrichment of proteins between Fumarate-alkyne labeling group and control group by PBSP. Succinated secreted proteins are highlighted in red. *p*-values were determined by two-sided Student *t*-tests. **c** Proportion of known secretory proteins among the succinated secreted proteins identified by PBSP. **d** KEGG pathway analysis of the succinated secreted proteins identified by PBSP with one-sided

*p*-values from Fisher's Exact test. **e** Overlap of succinated and itaconated secreted proteins identified by PBSP. **f** Volcano plot comparing protein enrichment under control vs. exosome-inhibited conditions (GW4869 treatment). Exosome-dependent succinated secreted proteins are highlighted in red. *p*-values were determined by two-sided Student *t*-tests. **g** Proportion of known exosome proteins within the exosome-dependent succinated secreted proteins. **h** KEGG analysis of the exosome-dependent succinated secreted proteins with one-sided *p*-values from Fisher's Exact test.

substrates, underscoring PTM-specific regulation. These datasets provide valuable resources for the investigation of PTM-mediated protein secretion and intercellular communications.

Among exosome-mediated itaconated proteins, we prioritized functional characterization of tyrosine kinase FYN. Biochemical validation pinpointed Cys239 within its SH2 domain as the primary itaconation site. This modification significantly suppressed inhibitory phosphorylation at Tyr531, indicative of enhanced kinase activation. Given FYN's role in phosphorylating diverse targets (including NRF2), macrophage-derived exosomes harboring itaconated FYN could potentially regulate recipient cell signaling—particularly within pathological contexts like tumor microenvironments. Further mechanistic studies are warranted to delineate how FYN itaconation orchestrates intercellular crosstalk.

We anticipate broad adoption of PBSP for profiling itaconation-dependent secretion across diverse cell types and physiological states. The platform's modular design also supports immediate extension to other PTMs. A key limitation of this study is that the chemical probes are structurally distinct from natural PTMs. This difference could alter their functionality and potentially interfere with the secretion process. Therefore, validation using orthogonal methods is recommended prior to any functional investigation. Another key current limitation is its inability to resolve intracellular PTM-mediated trafficking, as most

bioorthogonal probes lack subcellular targeting capability. Future iterations incorporating organelle-specific labeling strategies would empower unprecedented resolution of PTM-governed intracellular protein dynamics.

## Methods

### Animals
All animal experiments followed the National Institutes of Health Guide for the Care and Use of Laboratory Animals and were approved by the Center for New Drug Safety Evaluation and Research, China Pharmaceutical University (Nanjing, China) (CPUTML-ZST-20250820). All animals were housed in a specific pathogen-free (SPF) facility with 12-hour light/dark cycles for 7 days before the experiments.

### Cell culture
Raw264.7, HEK293T and iBMDM cells were obtained from ATCC. The IRG1-KO construct based on the Raw264.7 cell line had been generated by Chu Wang's lab at Peking University and was provided to us as a generous gift. All the cells were cultured in DMEM (Thermo Fisher Scientific) supplemented with 10% (vol/vol) fetal bovine serum (Thermo Fisher Scientific), 100 U/mL penicillin, and 100 µg/mL streptomycin at 37 °C under 5% $CO_2$.

## Antibodies and Reagents

ITalk was purchased from the ChomiX Biotech Co., Ltd. Nanjing, China (cat. no. CMX210035). Fumarate-Alkyne was purchased from Leyan (cat. no. 1595011). HNE-Alkyne and 4-OI were kindly gifted from the Chu Wang lab at Peking University. 1,6-Pr$_2$GalNAz was kindly gifted from the Xing Chen lab at Peking University. GW4869 was purchased from Selleck Chemicals (cat. no. S7609). The antibodies used for immunoblotting were mouse monoclonal anti-FYN antibody (cat. no. 200603, Zenbio), rabbit monoclonal anti-FLT-1 antibody (cat. no. R26069, Zenbio), rabbit monoclonal anti-Phospho-FYN (Tyr530) antibody (cat. no. 370334, Zenbio), rabbit monoclonal anti-Myc-Tag antibody (cat. no. AE070, Abclonal), rabbit monoclonal anti His-tag antibody (cat. no. AE086, Abclonal), HRP-labeled Streptavidin (cat. no. A0305, Beyotime), Goat Anti-Mouse HRP polyclonal antibody (cat. no. FDM007, Hang Zhou Fude Biological Technology), Goat Anti-Rabbit HRP polyclonal antibody (cat. no. FDR007, Hang Zhou Fude Biological Technology).

## Plasmids and site-directed mutagenesis

pECMV-Fyn-m-FLAG was purchased from Miaoling Plasmid (cat. no. P6464). For protein purification, Fyn was inserted into pET28a vector. For transfection, mutants were cloned from pECMV-Fyn-m-FLAG. Polymerase chain reactions (PCRs) were performed with Phanta DNA Polymerase purchased from Vazyme (cat. no. P520). Ligase-free cloning reactions were performed with Basic Seamless Cloning and Assembly Kit purchased from TransGen (cat. no. CU201). Plasmid products were transformed into DH5α competent cells purchased from Tsingke (cat. no. TSC-C14). DNA sequences were confirmed by Sanger sequencing before use.

Site-directed mutagenesis was used to introduce cysteine-to-serine substitutions into FYN. Polymerase chain reactions (PCRs) were performed with Phanta DNA Polymerase purchased from Vazyme (cat. no. P520). Ligase-free cloning reactions were performed with Basic Seamless Cloning and Assembly Kit purchased from TransGen (cat. no. CU201). Plasmid products were transformed into DH5α competent cells purchased from Tsingke (cat. no. TSC-C14). DNA sequences were confirmed by Sanger sequencing before use. The sequences of primers used for constructing the plasmids are provided in Supplementary Table 6.

## Protein purification

FYN was produced in *E. coli* BL21 (DE3) cells with an LB medium. Cells were grown at 37 °C until optical density at 600 nm (OD600) of 0.6–0.8. The temperature was then reduced to 16 °C and the cultures were grown for approximately 16 hours before harvesting. The cells were lysed in 20 mM Tris-HCl 8.0, 500 mM NaCl, 5% glycerol and 5 mM imidazole by sonication, the lysate was centrifuged at 20,000 g for 30 min at 4 °C, the resulting supernatant was filtrated by 0.22 μm filter and then applied onto a 3 mL Ni NTA Beads 6FF column (Lablead). Bound protein was eluted with a gradient of imidazole from 50 to 500 mM. Fractions containing FYN were ultrafiltered (concentrated) and buffer-exchanged into PBS, then stored at −80 °C.

## FYN recombinant protein labeling by ITalk

For concentration-dependent experiment, purified FYN recombinant protein (20 μM) was incubated with different concentrations of ITalk (from 0.2 mM to 5 mM) for 1 hour at 37 °C. The excess reagents were removed by methanol and chloroform (4:1) precipitation and protein samples were centrifuged at 8000 g for 5 min at 4 °C and washed twice with 1 mL cold methanol.

For iodoacetamide competitive experiment, purified FYN recombinant protein (20 μM) was incubated with different concentrations of iodoacetamide for 30 min at 37 °C, then samples were incubated with 1 mM ITalk for 1 hour at 37 °C. The excess reagents were removed by methanol and chloroform (4:1) precipitation and protein samples were centrifuged at 8000 g for 5 min at 4 °C and washed twice with 1 mL cold methanol.

Samples were redissolved by 0.4% SDS-PBS. 50 μL of sample was mixed with 0.5 mM CuSO4, 1 mM BTTAA, 100 μM Rhodamine-Azide, and 2.5 mM freshly prepared sodium ascorbate for 2 hours at 25 °C. The excess reagents were removed by methanol and chloroform (4:1) precipitation and samples were resolved on a 4–20% SDS-PAGE gel and imaged by Tanon 5200multi.

## Quantification of intracellular itaconate

Raw264.7 cells were plated at $2 \times 10^6$ per well in 6-well plates overnight. At the time 0 hour, cells were treated with 1 mM ITalk or DMSO for 2 hours in serum-free medium, respectively. Then the cells were gently washed three times with PBS in the plates and incubated with serum-free medium for 12 hours. Another well was treated with 100 ng/mL LPS for 14 hours. The cells in 0 hour, 2 hours and 14 hours were washed with PBS and collected by centrifugation. The cells were further washed three times with PBS by centrifugation. The cell pellets were lysed by sonication in ice-cold PBS containing 0.1% TritonX-100, centrifuged at 20,000 g for 30 min to remove cell debris, and protein concentrations were determined by BCA protein assay. After normalizing the protein concentration to 2 mg/mL of 100 μL, 900 μL cold methanol with 1 μM $^{13}$C-itaconate was added to extract the small molecule metabolites on ice. The mixture was incubated at −20 °C for 2 hours and was centrifuged at 20,000 g for 1 hour at 4 °C. The supernatant was collected and dried in a vacuum centrifuge and resolve in 50 μL water and analyzed by LC-SRM. The LC-SRM system is composed of an AB SCIEX 5500 triple-quadrupole mass spectrometer and a SHIMADZU DGU-20A liquid chromatography instrument with an Agilent column. The buffer gradient is 100%-0 Buffer A (100% water, 0.1% formic acid) and 0–100% Buffer B (100% methanol, 0.1% formic acid) for 10 min. The absolute concentration of itaconate was calculated according to the standard curve of itaconate.

## In situ proteins labeling by distinct probes

The Raw264.7 cells were grown to 80% confluence. The cells were treated with DMSO or different concentration of probes for different time (For ITalk, the concentration ranged from 0.5 mM to 5 mM, time ranged from 1 hour to 4 hours, for Fumarate-Alkyne, the concentration was 2 mM and time was 6 hours, for HNE-Alkyne and 1,6-Pr$_2$GalNAz the concentration was 100 μM and time was 6 hours). The cells were washed three times with PBS and centrifuged at 1,000 rpm for 3 min. The cells were stored at −80 °C. The cells were lysed in 1 mL RIPA lysis buffer (50 mM Tris pH 8, 150 mM NaCl, 0.1% SDS, 0.5% sodium deoxycholate, 1% Triton X-100) containing 1 × protease inhibitor cocktail (Roche) with sonication. The cell lysates were collected by centrifugation (20,000 g, 30 min) at 4 °C to remove debris. The protein concentration was determined by using the BCA protein assay kit. For visualizing the probes labeling efficiency by in-gel fluorescence, 50 μL of cell lysates (2 mg/mL) was mixed with 0.5 mM CuSO$_4$, 1 mM BTTAA, 100 μM Rhodamine-Azide (for streptavidin blotting, the probe was Biotin-Azide and for 1,6-Pr$_2$GalNAz the probe was Biotin-Alkyne), and 2.5 mM freshly prepared sodium ascorbate for 2 hours at 25 °C. The excess reagents were removed by methanol and chloroform (4:1) precipitation and samples were resolved on a 4–20% SDS-PAGE gel and for In-gel fluorescence, gels were imaged by Tanon 5200multi, for streptavidin blotting, after SDS-PAGE, the gels were transferred to a PVDF membrane, the blots were then blocked in 5% (w/v) milk in TBS-T (Tris-buffered saline, 0.1% Tween 20) for 1 hour at room temperature, the blots were stained with 0.3 μg/mL HRP-labeled Streptavidin in TBS-T for 1 hour at room temperature and then washed three times with TBS-T for 5 min each time before to development with Clarity Western ECL Blotting Substrates (Thermo Fisher) and imaging on the Tanon 5200multi. The gels were stained by Coomassie brilliant blue to demonstrate equal loading.

## Labeling of secreted proteins by distinct PTM probes

Raw264.7 cells were grown to 80% confluence and treated with either DMSO or distinct probes at specified concentrations and durations: ITalk (0.5–5 mM, 1–4 hours), Fumarate-Alkyne (2 mM, 6 hours), or HNE-Alkyne/1,6-Pr$_2$GalNAz (100 µM, 6 hours). Cells were washed thrice with PBS, incubated in serum-free medium for 12 hours, and conditioned medium was collected. Proteins were concentrated using 10 kDa filters (Millipore, UFC801024), resuspended in 100 µL PBS with 0.4% SDS, and quantified via BCA assay. To assess labeling efficiency, 50 µL secreted proteins (2 mg/mL) underwent click chemistry with 0.5 mM CuSO$_4$, 1 mM BTTAA, 100 µM Rhodamine-Azide (for in-gel fluorescence) or Biotin-Azide (for streptavidin blotting; Biotin-Alkyne for 1,6-Pr$_2$GalNAz samples), and 2.5 mM sodium ascorbate (2 hours, 25 °C). Excess reagents were removed by methanol:chloroform (4:1) precipitation. Samples were resolved on 4%–20% SDS-PAGE gels: for in-gel fluorescence, gels were imaged (Tanon 5200multi); for streptavidin blotting, proteins were transferred to PVDF, blocked in 5% milk/TBS-T (1 hour, RT), incubated with 0.3 µg/mL HRP-streptavidin (1 hour, RT), washed thrice with TBS-T, developed with Clarity Western ECL, and imaged. Equal loading was confirmed by Coomassie brilliant blue staining.

## Enrichment of labeled proteins by ITalk

To validate novel itaconated proteins, Raw264.7 cells were grown to 80% confluence and treated with DMSO or 1 mM ITalk for 2 hours. For FYN mutation studies, HEK293T cells were grown to 70% confluence in 10 cm dishes and transiently transfected with FYN mutants for 16 hours, followed by treatment with 1 mM ITalk or DMSO for 6 hours. Cells were washed thrice with PBS and centrifuged at 1000 rpm (approx. 200 g) for 3 min. Cell pellets were lysed in 1 mL RIPA buffer (50 mM Tris pH 8, 150 mM NaCl, 0.1% SDS, 0.5% sodium deoxycholate, 1% Triton X-100) containing 1× protease inhibitor cocktail (Roche) using sonication. Lysates were clarified by centrifugation (20,000 g, 30 min, 4 °C), and protein concentration was determined by BCA assay. For click chemistry, 1 mL lysate (2 mg/mL) was reacted with 0.5 mM CuSO$_4$, 1 mM BTTAA, 100 µM Biotin-Azide, and 2.5 mM sodium ascorbate (2 hours, 25 °C). Excess reagents were removed by methanol:chloroform (4:1) precipitation; samples were centrifuged (8000 g, 5 min, 4 °C) and washed twice with cold methanol.

Precipitated proteins were redissolved in 1 mL RIPA buffer. For each sample, 200 µL streptavidin magnetic beads were equilibrated with RIPA buffer (2 × 1 mL washes), then incubated with 2 mg protein in 1 mL RIPA buffer overnight at 4 °C. Unbound lysates were stored at −20 °C for Western blotting. Post-enrichment, beads were collected magnetically and washed sequentially with: RIPA buffer (2×1 mL), 1 M KCl (1 × 1 mL), 0.1 M Na$_2$CO$_3$ (1 × 1 mL), 2 M urea/10 mM Tris-HCl pH 8.0 (1 × 1 mL), and RIPA buffer (2 × 1 mL). Enriched proteins were eluted by boiling beads in 30 µL 3× protein loading buffer containing 2 mM biotin and 20 mM DTT (95 °C, 10 min). Samples were resolved on 4–20% SDS-PAGE gels, transferred to PVDF membranes, blocked with 5% milk/TBS-T (1 hour, RT), and probed with anti-FYN (1:1,000) and anti-FLT-1 (1:1,000) antibodies in TBS-T overnight at 4 °C. Membranes were washed (3 × 5 min TBS-T), incubated with HRP-conjugated Goat Anti-Mouse (1:10,000) and Goat Anti-Rabbit (1:10,000) antibodies (1 hour, RT), washed again (3 × 5 min TBS-T), developed with Clarity Western ECL, and imaged (Tanon 5200multi).

## Identification of itaconated sites on FYN

The concentration of purified FYN protein was diluted to 0.5 µg/µL with PBS (pH 7.4). The diluted protein (100 µL per reaction in 1.5 mL tube) was incubated with 5 mM itaconate for 2 hours at 37 °C. The resulting sample was transferred onto the membrane of 10 k ultracentrifuge filter (Sartorius) and centrifuged at 12,000 g for 15 min, RT. The buffer was changed to 100 µL 8 M urea/PBS containing 10 mM DTT

for 30 min at 37 °C, followed by addition of 20 mM iodoacetamide for another 30 min at 35 °C. Then the membrane was washed with 200 µL of 50 mM ammonium bicarbonate for five times, filled with 200 µL of 50 mM ammonium bicarbonate containing 10 ng/µL trypsin for 15 hours at 37 °C. The peptides were collected by centrifugation at 12,000 g for 15 min. The membrane was washed twice with 200 µL of 50 mM ammonium bicarbonate. The sample was dried out using a SpeedVac (RT), dissolved with 40 µL of 0.1% formic acid and subjected to LC-MS/MS analysis. The identification of itaconation sites was performed by pFind with the following key parameters: MS Instrument, HCD-FTMS; using the proper database of different species; Enzyme, C terminal of KR; unselect Open Search; Variable modification, Carbamidomethyl [C], Oxidation [M], and ITA [C]. ITA[C] was configured using pConfig (the Composition was C(5)H(6)O(4), the Mass was 130.026604 and Site was C). No additional parameter changes were applied.

## Detection of FYN Tyr531 phosphorylation

The Raw264.7 cells or IRG1-KO Raw264.7 cells were grown to 80% confluence in 10 cm dishes. The cells were treated with 3 mM itaconate or 500 µM 4-OI for 24 hours, For LPS treated experiment, cells were treated with 100 ng/mL LPS for different time. Cells were harvested and lysed in 1 mL RIPA buffer (50 mM Tris pH 8, 150 mM NaCl, 0.1% SDS, 0.5% sodium deoxycholate, 1% Triton X-100) using sonication. Lysates were clarified by centrifugation (20,000 g, 30 min, 4 °C). Protein concentration was determined via BCA assay. Samples were resolved on 4%–20% SDS-PAGE gels and transferred to PVDF membranes. Membranes were blocked with 5% non-fat milk in TBS-T (Tris-buffered saline, 0.1% Tween 20) for 1 hour at room temperature, then probed with anti-Phospho-FYN (Tyr531) antibody (1:1,000) in TBS-T overnight at 4 °C. After three 5 min TBS-T washes, membranes were incubated with Goat Anti-Rabbit HRP (1:10,000) for 1 hour at room temperature, washed again (3 × 5 min TBS-T), developed with Clarity Western ECL Substrates (Thermo Fisher), and imaged (Tanon 5200multi).

## Assessment of Fyn phosphorylation in BMDMs upon stimulation with itaconate derivatives and LPS

Male C57BL/6 J (6-8 week-old), were sourced from Jiangsu Jicui Pharmaceutical & Biotechnology Co., Ltd. Femurs and tibias were aseptically harvested from 6-8-week-old C57BL/6 mice. Bone marrow cells were flushed out with sterile PBS containing 2% FBS and passed through a 70 µm cell strainer to obtain a single-cell suspension. Cells were resuspended in high-glucose DMEM supplemented with 10% (vol/vol) heat-inactivated FBS, 1% penicillin-streptomycin, and 10 ng/mL recombinant murine macrophage colony stimulating factor (M-CSF, PeproTech, Cat. no. 315-02). Cells were seeded into non-tissue-culture-treated Petri dishes and maintained at 37 °C in a humidified 5% CO$_2$ incubator. Fresh M-CSF containing medium was replenished on day 3. By day 6, adherent cells displayed characteristic macrophage morphology and were confirmed as differentiated BMDMs. BMDMs were treated with 4-OI (250 µM or 500 µM) or ITA (3 mM or 6 mM) for 24 hours. For LPS stimulation, cells were exposed to 100 ng/mL LPS for 6 hours. Cells were then harvested and lysed in RIPA buffer supplemented with protease and phosphatase inhibitor cocktails. Lysates were clarified by centrifugation at 12,500 rpm for 10 min at 4 °C. Proteins were quantified by BCA assay, resolved on SDS-PAGE, and analyzed by immunoblotting with antibodies against FYN (1:1000) and phospho-FYN (Tyr531) (1:1000). Signals were detected by enhanced chemiluminescence and imaged with a Tanon 5200multi system.

## Detection of FYN in exosomes

Raw264.7 cells were grown to 80% confluence in 15 cm dishes, washed three times with PBS, and incubated in serum-free medium

for 12 hours. Conditioned medium from three dishes was collected and centrifuged sequentially: first at 500 g for 10 min (4 °C, twice) to remove suspended cells, then at 2000 g for 10 min (4 °C, twice) to pellet debris. Ectosomes were collected by centrifugation at 12,000 g for 40 min (4 °C). The supernatant was filtered through a 0.22 μm membrane and ultracentrifuged at 120,000 g for 70 min (4 °C). After supernatant removal, the pellet was resuspended in PBS and re-centrifuged at 120,000 g for 70 min (4 °C) to isolate exosomes. Purified exosomes were resuspended in 100 μL PBS. For detection, 50 μL of exosome sample was mixed with 50 μL RIPA lysis buffer and sonicated. Samples were resolved on 4–20% SDS-PAGE gels, transferred to PVDF membranes, blocked with 5% milk in TBS-T (Tris-buffered saline with 0.1% Tween 20) for 1 hour at room temperature, and probed with anti-FYN antibody (1:1,000 dilution) in TBS-T overnight at 4 °C. Membranes were washed three times with TBS-T (5 min per wash), incubated with HRP-conjugated Goat Anti-Mouse secondary antibody (1:10,000 dilution) for 1 hour at room temperature, washed again three times with TBS-T (5 min per wash), developed using Clarity Western ECL Substrates (Thermo Fisher), and imaged on a Tanon 5200multi system.

### Quantitative profiling of itaconated and succinated secretomes by ITalk and fumarate-alkyne

The Raw264.7 cells were grown to 80% confluence in 10 cm dishes. The cells were treated with 1 mM ITalk (or fumarate-alkyne) or DMSO for 2 hours. For exosome inhibition status labeling, cells were treated with 1 mM ITalk (or fumarate-alkyne) and 5 μM GW4869 for 2 hours. The cells were washed three times with PBS and incubated with serum-free medium with GW4869 or DMSO for 12 hours. Conditioned medium was collected and concentrated using 10 kDa filter tubes (Millipore, UFC801024). The concentrated proteins were resuspended in 100 μL of 0.4% SDS-PBS buffer. The protein concentration was determined by using the BCA protein assay kit. 50 μL of conditioned medium (2 mg/mL) was mixed with 0.5 mM CuSO4, 1 mM BTTAA, 100 μM azide-biotin, and 2.5 mM freshly prepared sodium ascorbate for 2 hours at 25 °C. The excess reagents were removed by methanol and chloroform (4:1) precipitation and protein samples were centrifuged at 8000 g for 5 min at 4 °C and washed twice with 1 mL cold methanol.

One plug of the C18 SPE disc was inserted into a standard 200 μL tip, then 15 mg of C18 packing material was dissolved in acetonitrile and added into tip. 60 μL of methanol was added and centrifuged at 3000 g for 3 min for conditioning and 60 μL of 2% SDS was added and centrifuged at 3000 g for 3 min to blocking C18. Streptavidin beads were washed twice with 1 mL RIPA lysis buffer and then resuspend with RIPA buffer. The pre-washed Streptavidin beads slurry was loaded and centrifuged at 1000 g for 3 min. 60 μg secreted proteins were then loaded onto the tip and centrifuged at 100 g for 1 hour. After biotinylated proteins were captured, the tips were washed 4 times with 60 μL of RIPA buffer at 1500 g for 1 min. Then C18 was activated with 60 μL of 0.5%(v/v) acetic acid in 80% acetonitrile by centrifuged at 6000 g for 3 min. For reduction of cysteine residues, 20 μL of 50 mM ammonium bicarbonate containing 10 mM DTT was added and incubated at 25 °C with 600 rpm for 15 min, the solution was removed with centrifugation at 6,000 g for 3 min. For digestion, 10 μL 50 mM ammonium bicarbonate containing 0.25 μg/μL trypsin, 50 mM IAA was loaded onto the tips and the samples were incubated at 37 °C in the dark for 1 hour with 600 rpm shaking to digest and alkylate the proteins. The digested peptides (retained on C18) were washed with 60 μL 10 mM pH = 10 ammonium bicarbonate for three times. Afterwards, the peptides were eluted with 60 μL of 0.5% (v/v) acetic acid in 80% acetonitrile at 5000 g for 5 min three times. Samples were dried in a vacuum centrifuge and stored at −80 °C until analysis.

### LC-MS/MS analysis

Peptides were separated using a loading column (100 μm × 2 cm) and a C18 separating capillary column (100 μm × 15 cm) packed in-house with Luna 3 μm C18(2) bulk packing material (Phenomenex, USA). The mobile phases (A: water with 0.1% formic acid and B: 94% acetonitrile with 0.1% formic acid) were driven and controlled by a Vanquish™ Neo UHPLC system (Thermo Fisher Scientific). The LC gradient for protein samples was held at 4% for the first 4 min of the analysis, followed by an increase from 5 to 20% B from 4 to 109 min, an increase from 20% to 35% B from 109 to 150 min and an increase from 35 to 99% B from 150 to 159 min. For the samples analyzed by Q Exactive-plus series Orbitrap mass spectrometers (Thermo Fisher Scientific), the precursors were ionized using an EASY-Spray ionization source (Thermo Fisher Scientific) source held at +2.0 kV compared to ground, and the inlet capillary temperature was held at 320 °C. Survey scans of peptide precursors were collected in the Orbitrap from 350 to 1800 Th with an AGC target of 3,000,000, a maximum injection time of 20 ms and a resolution of 70,000. The data-independent acquisition mode was selected. For each DIA window, resolution was set to 17,500. AGC target value for fragment spectra was set at 1,000,000 with an auto IT. Normalized CE was set at 28%. Default charge was 3 and the fixed first mass was set to 200 Th.

### Data analysis

The raw data were processed using DIA-NN in an advanced library-free module. The main search settings for in silico library generation were set as following: trypsin/P with maximum 3 missed cleavage; protein N-terminal M excision on; carbamidomethyl on C as fixed modification; oxidation on M as variable modification; peptide length from 7 to 30; precursor charge 1–4; precursor m/z from 300 to 1800; fragment m/z from 200 to 1800. The Mouse UniProt isoform sequence database (3AUP000000589) was used to annotate proteins for mouse samples. Other search parameters were set as following: quantification strategy was set to "QuantUMS (high precision)" mode; cross-run normalization was off; MS2 and MS1 mass accuracies were set to 0, allowing the DIA-NN to automatically determine mass tolerances; Scan window was set to 0 corresponding to the approximate average number of data points per peak; Peptidoforms and MBR were turned on; neural network classifier was single-pass mode. The enrichment ratios were calculated using raw intensity values.

### Statistical analysis

Gene ontology and KEGG analysis were conducted using the DAVID, Exosome-dependent data was obtained from ExoCarta. The heatmap, bar and line charts in the figures were generated using GraphPad Prism 9. The schematic workflows were created in https://BioRender.com. Significance was defined as a $*p < 0.05$, $**p < 0.01$ and $***p < 0.001$ for the two-tailed Student's t-test. Results are expressed as mean ± s.d. Fold change in relation to control groups of three independent cell culture and subsequent procedures.

## Data availability

The data that support the findings of this study are available in the supplementary material of this article. Source data are provided with this paper.

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

## Acknowledgements

This work was supported by the National Key Research and Development Program of China (No. 2024YFA1308000, W.Q.), National Natural Science Foundation of China (22477066 and 92478128 to W.Q.; 22207126 to Y.Z.; 32301241 to S.Z.), Youth Talent Cultivation Fund of Tsinghua University (W.Q.), "Dushi Plan" from Tsinghua University (W.Q.), Beijing Frontier Research Center for Biological Structure, the Fundamental Research Funds from Beijing National Laboratory for Molecular Sciences (BNLMS202301, W.Q.) and the Shenzhen Medical Research Fund (B2401004, W.Q.). W.Q. is supported by Bayer Investigator Award. We thank Prof. Chu Wang from Peking University for providing of IRG1-KO cell line and HNE probe. We thank Prof. Xing Chen from Peking University for providing of 1,6-Pr$_2$GalNAz probe.

## Author contributions

W.L., Y.Z., S.Z. and W.Q. designed the research and analyzed all the data except where noted. W.L. and Y.Z. performed all experiments except where noted. W.L., Y.Z. and W.Q. designed the proteomics experiments and analyzed the MS data. P.W., X. N. and S. Z. performed the FYN phosphorylation experiments. W.L., Y.Z. and W.Q. wrote the paper with input from all authors.

## Competing interests

The authors declare no competing interests.
