## [Transparent Peer review file · Nature Communications]

Spatiotemporal profiling of modification-specific proteome secretion uncovers an itaconation-activated tyrosine kinase

Corresponding Author: Professor Wei Qin

Version 0:

Reviewer comments:

Reviewer #1

(Remarks to the Author)

In this manuscript, Qin and colleagues developed PTM-TransitID, a novel approach for identifying secreted proteins bearing specific post-translational modifications (PTMs). They applied this method to establish a comprehensive profile of macrophage-secreted proteins that were modified by itaconation or succination in exosomes. The optimized workflow integrated a FISAP-based sample preparation step with DIA-based mass spectrometry, and overcome the challenges associated with limited secreted protein quantities while achieving high sensitivity and exceptional coverage. Notably, this approach led to the discovery that itaconation of Cys239 in the tyrosine kinase FYN enhances its kinase activity, suggesting a potential role in regulating intercellular communication.

Overall, this is a well-written and rigorously conducted manuscript that makes an important contribution to the fields of PTM chemical biology and protein secretion. The method could be broadly applied to investigate diverse types of PTMs as well as their functional roles in intercellular communication and immune responses. I support its publication in Nature Communications after the following concerns are addressed.:

Major comments:

1. The authors mutated each cysteine residue in FYN and found that only the Cys239 mutation reduced ITalk labeling. To meet established standards for confirming the PTM modification site, the MS/MS spectra to validate this modification site should be provided.
2. While PTM-TransitID can specifically detect secretion of those PTM-bearing proteoforms, evaluating the impact of such PTMs on the secretion efficiency requires further comparative analysis. The authors should quantify the global proteome secretion (e.g., via TurboID pulse-chase) and compare the secretion rates between proteins with specific PTMs and their unmodified counterparts.

Minor Comments:

1. The name "PTM-TransitID" is potentially misleading, as the methodology is unrelated to proximity labeling or TransitID technology. Renaming the approach is strongly recommended to avoid confusion.
2. Figure panels 3c, 3d, 4e, 6d, and 6h: The dot sizes represent protein counts, but exact numerical values should be clearly labeled within or adjacent to the plots.

Reviewer #2

(Remarks to the Author)

Building on concepts of the proximity labeling approach TransitID, Lu et al. present an elegant novel method for PTM-directed identification of secreted proteins. By combining in cellulo labelling with bioorthogonal probes and click chemistry, their approach enables streptavidin-based enrichment of proteins modified by specific PTMs, depending on the probe used. The authors generate extensive datasets for itaconation and succination, and provide proof-of-concept data for glycosylation

and carbonylation. Applying their workflow, they identify exosome-dependent itaconated secreted proteins using an exosome inhibitor, and further explore itaconation of FYN and its impact on kinase activity.

With growing interest in spatial intra- and intercellular biology for various disease and developmental settings, there remains a pressing need for methodologies that enable studying cellular communication. Lu et al. address this gap and offer a new approach to study the biochemical language of cell-to-cell signaling. With technological and analytical advances such as DIA and FISAP, this field of study is expected to continue in that direction.

However, as currently presented, the manuscript would require revisions for publication in Nature Communications.

Major concerns:

1. Currently, the data are generated using only one macrophage cell line. The results would be further strengthened by demonstrating the workflow in at least one additional cell line to support broad applicability of the approach. This could include macrophage or non-macrophage cell types, as the focus of the study is primarily on the technological advancement rather than a specific biological context.
2. How do the authors ensure that the identified proteins are modified by the PTM under endogenous conditions? Excessive supplementation of the chemical probe may artificially drive protein modification in a concentration-dependent manner, questioning the biological relevance. The overlap with other datasets, as shown in Figure 3, does increase confidence in the method, however, both of these datasets are also probe-based and originate from the same research group, potentially introducing shared methodological biases. Validation using an orthogonal method would further improve the manuscript.
3. The experimental design of the LC-MS/MS setups is somewhat unclear to me. Were the control conditions also performed in triplicate? If so, it is unclear why statistical analyses were only applied to the exosome-related but not to the prior PTM-TransitID experiments? Given the quantitative nature of the DIA data, I strongly recommend that appropriate statistical analyses are included for all PTM-TransitID experiments. How is the enrichment ratio (\log_2FC) currently calculated? Are raw intensities used, or are values normalized (LFQ, iBAQ, or other methods)? I would advise the authors to expand and clarify the data analysis section in the Materials and Methods to address these points in more detail.

Minor concerns:

1. What is the rationale for the chosen durations of the pulse and chase periods? This information should be added to the manuscript. Similarly, why is serum-free medium used during the chase phase? While serum-free culturing is frequently performed in EV research, recent studies (e.g. PMID 39842778 and PMID 33397101) have shown that serum deprivation can alter the secreted proteome. Can the authors clarify why serum-deprivation is preferred for the PTM-TransitID approach? Have the authors tested the approach without serum-deprivation? Since serum deprivation can include changes in secretion profiles, requiring this step may limit the applicability of the approach in certain biological settings.
2. There seems to be a duplicated sentence at lines 213-216.
3. Could the authors provide the complete dataset of all identified and quantified proteins in the Supplementary Tables, rather than only the enrichment ratios for proteins retained after filtering with other datasets?
4. Can panels b and d of Figure 1 be moved to Supplementary information? Figure 1 seems currently slightly cluttered, while this is arguably the Figure that will be most watched by the readers to comprehend the method. Additionally, in Figure 1a the different proteins are currently rather large relative to the cell, more closely resembling organelles.
5. Some bar charts, such as Figure 3e and Figure S3, display the wrong value (mean + SD) rather than just the mean, while others, such as Figures 3f-g, present the mean. I recommend removing the numerical values from all bar charts, as they do not improve readability in my opinion. Alternatively, the authors could at least ensure consistency in how values are presented across all figures. Additionally, the figure captions should specify what the error bars represent, and the number of replicates (N) should be clearly stated.
6. The study (perhaps unintentionally) also highlights a current challenge in precisely identifying which amino acid residues are carrying the PTM. Have the authors considered performing click chemistry and streptavidin enrichment after trypsin digestion to directly pinpoint PTM-carrying peptides? This could reduce the need for extensive mutation studies, enabling more targeted mutation efforts instead. While this is not required for the publication of the current manuscript, it could be an interesting avenue for the authors to explore in future work.

Reviewer #3

(Remarks to the Author)

The manuscript "Spatiotemporal profiling of modification-specific proteome secretion uncovers an itaconation-activated tyrosine kinase" by Lu and colleagues introduces PTM-TransitID which is a chemoproteomic technique that uses bioorthogonal probes and advanced mass spectrometry to identify secreted, PTM-bearing proteins. The authors used this technique to identify itaconated proteins and discovered itaconation of tyrosine kinase FYN that enhances its activity.

This study is well executed and offers insights into the role of itaconate in protein modification. However, the current manuscript does not convincingly demonstrate whether itaconate drives the observed effects or if they are predominantly mediated by octyl-itaconate (probes are based on OI rather than itaconate), which possesses distinct biological activities compared to itaconate. Further data and clarifications are necessary to fully validate the findings.

1. The authors used ITalk (previously characterized in their earlier publication (18)) which is based on octyl-itaconate (OI). This distinction should be clearly stated and discussed, and the title might be adjusted accordingly. Since OI and itaconate are metabolically and functionally distinct (with different electrophilic properties), the authors should carefully clarify these differences and appropriately temper their interpretations and conclusions.

2. The experimental model relies on RAW264.7 macrophage-like cells which is a proliferative cell line. To enhance the physiological relevance of key findings, particularly those involving the kinase FYN, some key findings should be repeated in primary macrophages. Some assays were conducted in HEK293T cells (Figure 5) but these key findings were not repeated in RAW264.7 cells.

3. RAW264.7 cells are highly responsive to external stimuli which can induce endogenous itaconate production. The authors should clarify whether endogenous itaconate synthesis occurs under their experimental conditions in RAW264.7 cells or whether itaconate is derived from ITalk. The authors might consider employing cells deficient in IRG1 or implementing metabolic tracing approaches.

4. Figure 5f depicts effects of exogenous itaconate treatment on FYN. Does endogenously synthesized itaconate, for instance following LPS stimulation, produce comparable effects?

Version 1:

Reviewer comments:

Reviewer #1

(Remarks to the Author)

The authors have a good job in addressing the suggestions and concerns from the reviewers and I support its publication

Reviewer #2

(Remarks to the Author)

I commend the authors for their thorough revisions. My concerns have been fully addressed, and I now support publication of the manuscript in Nature Communications. I believe PBSP has potential for broad application within the field.

Reviewer #3

(Remarks to the Author)

The reviewers have addressed my concerns. The manuscript now includes data on itaconate and its impact on protein phosphorylation.

The authors also include RAW264.7 IRG1-KO cells. However, I cannot find any additional details on this cell line, including how it was engineered.

Below are our point-by-point responses to the questions and concerns raised by the reviewers (colored in blue). The newly added or revised texts have also been colored in yellow in the manuscript files to facilitate the second round of review.

Reviewer #1 (Remarks to the Author):

In this manuscript, Qin and colleagues developed PTM-TransitID, a novel approach that achieves high sensitivity and exceptional coverage. Notably, this approach led to the discovery that itaconation of Cys239 in the tyrosine kinase FYN enhances its kinase activity, suggesting a potential role in regulating intercellular communication.

Overall, this is a well-written and rigorously conducted manuscript that makes an important contribution to the fields of PTM chemical biology and protein secretion. The method could be broadly applied to achieve for identifying secreted proteins bearing specific post-translational modifications (PTMs). They applied this method to establish a comprehensive profile of macrophage-secreted proteins that were modified by itaconation or succination in exosomes. The optimized workflow integrated a FISAP-based sample preparation step with DIA-based mass spectrometry, and overcame the challenges associated with limited secreted protein investigation of diverse types of PTMs as well as their functional roles in intercellular communication and immune responses. I support its publication in Nature Communications after the following concerns are addressed.:

Response: We thank the reviewer for the positive comments.

Major comments:

1. The authors mutated each cysteine residue in FYN and found that only the Cys239 mutation reduced ITalk labeling. To meet established standards for confirming the PTM modification site, the MS/MS spectra to validate this modification site should be provided.

*Response: Thank you to the reviewer for this excellent suggestion. To address this point, we performed LC-MS/MS analysis on recombinant FYN that had been pre-treated with itaconate. This experiment identified five peptides modified by itaconation, including the peptide containing Cys239 (**Rebuttal Figure 1; Figure 5c and S5c in the revised manuscript**). However, because Cys239 and Cys240 are in close proximity, the MS/MS spectra could not unambiguously assign the modification to one residue over the other. This finding underscores the critical role of our subsequent site-directed mutagenesis experiments, which definitively identified Cys239 as the primary site of itaconation on FYN.*

Rebuttal Figure 1. Tandem MS spectra of peptides bearing itaconation on FYN.

2. While PTM-TransitID can specifically detect secretion of those PTM-bearing proteoforms, evaluating the impact of such PTMs on the secretion efficiency requires further comparative analysis. The authors should quantify the global proteome secretion (e.g., via TurboID pulse-chase) and compare the secretion rates between proteins with specific PTMs and their unmodified counterparts.

Response: Thank you for raising this important point. We followed the reviewer's suggestion and attempted to generate a Raw264.7 cell line stably expressing an ER-resident TurboID. However, as has been documented in previous studies¹, transgenic expression in macrophage cell lines is notoriously challenging. Despite our efforts, we were unable to establish a viable cell line that enabled efficient

detection of biotinylated proteins in the culture medium. Instead, we compared our itaconation-secretome to succination-secretome dataset. Our data reveal a critical finding: while itaconation and succination share similar chemical mechanisms and intracellular substrate profiles, the profiles of secreted proteins modified by each are strikingly different (**Figure 6e**), strongly suggesting that itaconation has a unique and specific impact on the secretome.

Minor Comments:

1. The name "PTM-TransitID" is potentially misleading, as the methodology is unrelated to proximity labeling or TransitID technology. Renaming the approach is strongly recommended to avoid confusion.

Response: Thank you for pointing this out. We have renamed this method as "PTM-based secretome profiling (PBSP)".

2. Figure panels 3c, 3d, 4e, 6d, and 6h: The dot sizes represent protein counts, but exact numerical values should be clearly labeled within or adjacent to the plots.

Response: Thank you for pointing this out. We have added the values accordingly.

Reviewer #2 (Remarks to the Author):

Building on concepts of the proximity labeling approach TransitID, Lu et al. present an elegant novel method for PTM-directed identification of secreted proteins. By combining in cellulo labelling with bioorthogonal probes and click chemistry, their approach enables streptavidin-based enrichment of proteins modified by specific PTMs, depending on the probe used. The authors generate extensive datasets for itaconation and succination, and provide proof-of-concept data for glycosylation and carbonylation. Applying their workflow, they identify exosome-dependent itaconated secreted proteins using an exosome inhibitor, and further explore itaconation of FYN and its impact on kinase activity.

With growing interest in spatial intra- and intercellular biology for various disease and developmental settings, there remains a pressing need for methodologies that enable studying cellular communication. Lu et al. address this gap and offer a new approach to study the biochemical language of cell-to-cell signaling. With technological and analytical advances such as DIA and FISAP, this field of study is expected to continue in that direction.

However, as currently presented, the manuscript would require revisions for publication in *Nature Communications*.

Response: We thank the reviewer for the positive comments.

Major concerns:

1. Currently, the data are generated using only one macrophage cell line. The results would be further strengthened by demonstrating the workflow in at least one additional cell line to support broad applicability of the approach. This could include macrophage or non-macrophage cell types, as the focus of the study is primarily on the technological advancement rather than a specific biological context.

Response: We thank the reviewer for the great suggestion. Accordingly, we performed PBSP profiling in immortalized bone marrow-derived macrophages (iBMDMs) under the same ITalk labeling conditions. Using identical filtering criteria, we identified 54 itaconated secreted proteins (**Rebuttal Figure 2a; Figure S3b in the revised manuscript**). Among these, 38 are known secreted proteins (70.4%) and 37 were previously reported as itaconated (68.5%), mirroring the specificity we observed in Raw264.7 cells (**Rebuttal Figure 2b, c; Figure S3c, d in the revised manuscript**). These results demonstrate that PBSP is broadly applicable and can reveal cell type-specific heterogeneities.

Rebuttal Figure 2. PBSP profiling of itaconated secreted proteins in iBMDM. a) Volcano plot showing the enrichment of proteins by PBSP in iBMDM cells. Itaconated secreted proteins are highlighted in red. b) Proportion of known secretory proteins among the itaconated secreted proteins identified by PBSP in Raw264.7 and iBMDM cells. c) Proportion of known itaconated proteins among the itaconated secreted proteins identified by PBSP in Raw264.7 and iBMDM cells.

2. How do the authors ensure that the identified proteins are modified by the PTM under endogenous conditions? Excessive supplementation of the chemical probe may artificially drive protein modification in a concentration-dependent manner, questioning the biological relevance. The overlap with other datasets, as shown in Figure 3, does increase confidence in the method, however, both of these datasets

are also probe-based and originate from the same research group, potentially introducing shared methodological biases. Validation using an orthogonal method would further improve the manuscript.

Response: We thank the reviewer for raising this important point. It is reasonable to question whether a chemical probe can accurately represent endogenous modification events, given that it is chemically distinct from the natural modification. However, tracking the dynamics of protein modifications requires the incorporation of a handle for detection, and we are not aware of any other viable, systematic methods to achieve this in a live-cell context. We would like to emphasize that this is a fundamental consideration—and potential limitation—shared by nearly all chemical biology probes for PTMs. That said, probes such as ITalk and FA-alkyne have been extensively validated in prior studies from other research groups²⁻⁴, which have demonstrated that they reliably label *bona fide* endogenous substrates using a physiologically relevant concentration. In direct response to the reviewer's comment, we performed LC-MS/MS analysis on recombinant FYN treated with natural itaconate. The results confirm that FYN is indeed modified by itaconation, thereby validating the labeling findings obtained with ITalk (**Rebuttal Figure 1; Figure 5c and S5c in the revised manuscript**). In the Discussion section of the revised manuscript, we have added a discussion on the potential influence of chemical probes on protein secretion and the importance of orthogonal validation in subsequent studies.

3. The experimental design of the LC-MS/MS setups is somewhat unclear to me. Were the control conditions also performed in triplicate? If so, it is unclear why statistical analyses were only applied to the exosome-related but not to the prior PTM-TransitID experiments? Given the quantitative nature of the DIA data, I strongly recommend that appropriate statistical analyses are included for all PTM-TransitID experiments. How is the enrichment ratio (log2FC) currently calculated? Are raw intensities used, or are values normalized (LFQ, iBAQ, or other methods)? I would advise the authors to expand and clarify the data analysis section in the Materials and Methods to address these points in more detail.

Response: We appreciate the reviewer for raising this important point. In our original manuscript, the control conditions were indeed performed in triplicate. While our initial analysis followed an established pipeline from our previous work that did not require statistical comparison of controls^{5,6}, we have now performed these statistical comparisons in accordance with the reviewer's suggestion. Accordingly, we have re-analyzed the proteomic data and updated the relevant figures (**Figure 2b and 6b**). We note that this change did not significantly alter the definitive list of itaconate-modified proteins or any of the study's conclusions. Furthermore, we have clarified in the Methods section that enrichment ratios were calculated using raw intensity values from DIA-NN.

Minor concerns:

1. What is the rationale for the chosen durations of the pulse and chase periods? This information should be added to the manuscript. Similarly, why is serum-free medium used during the chase phase? While

serum-free culturing is frequently performed in EV research, recent studies (e.g. PMID 39842778 and PMID 33397101) have shown that serum deprivation can alter the secreted proteome. Can the authors clarify why serum-deprivation is preferred for the PTM-TransitID approach? Have the authors tested the approach without serum-deprivation? Since serum deprivation can include changes in secretion profiles, requiring this step may limit the applicability of the approach in certain biological settings.

Response: We thank the reviewer for this insightful comment. As suggested, we have now included an explanation for the chosen experimental parameters. The 2-hour pulse duration was selected to achieve high temporal resolution. For the chase phase, the use of serum-free medium follows established TurboID-based pulse-chase protocols⁷. We evaluated serum-containing medium but found that the high abundance of proteins in the serum non-specifically bound to the beads, severely interfering with the enrichment of labeled proteins. This rationale has been added to the revised manuscript.

2. There seems to be a duplicated sentence at lines 213-216.

Response: We have corrected this mistake.

3. Could the authors provide the complete dataset of all identified and quantified proteins in the Supplementary Tables, rather than only the enrichment ratios for proteins retained after filtering with other datasets?

Response: We have provided the complete dataset of all identified and quantified proteins in the Supplementary Tables.

4. Can pannels b and d of Figure 1 be moved to Supplementary informatio? Figure 1 seems currently slightly cluttered, while this is arguably the Figure that will be most watched by the readers to comprehend the method. Additionally, in Figure 1a the different proteins are currently rather large relative to the cell, more closely resembling organelles.

Response: We appreciate the reviewer's suggestion. Accordingly, we have moved these figure panels to the Supplementary Information and reduced the size of the protein representations.

5. Some bar charts, such as Figure 3e and Figure S3, display the wrong value (mean + SD) rather than just the mean, while others, such as Figures 3f-g, present the mean. I recommend removing the numerical values from all bar charts, as they do not improve readability in my opinion. Alternatively, the authors could at least ensure consistency in how values are presented across all figures. Aditionally, the figure

captions should specify what the error bars represent, and the number of replicates (N) should be clearly stated.

Response: We thank the reviewer for this observation. We have revised the figures to present all data consistently as Mean \pm SD. The figure legends have been updated to explicitly state that error bars represent the SD and to indicate the number of biological replicates (n) for each experiment.

6. The study (perhaps unintentionally) also highlights a current challenge in precisely identifying which amino acid residues are carrying the PTM. Have the authors considered performing click chemistry and streptavidin enrichment after trypsin digestion to directly pinpoint PTM-carrying peptides? This could reduce the need for extensive mutation studies, enabling more targeted mutation efforts instead. While this is not required for the publication of the current manuscript, it could be an interesting avenue for the authors to explore in future work.

Response: We thank the reviewer for this insightful suggestion. We agree that performing click chemistry post-digestion would be an excellent method for pinpointing the modified residues. However, this approach is technically challenging in our system due to the low abundance of itaconated proteins secreted into the medium. Identifying modification sites requires a larger amount of input material than is currently feasible for us to obtain. We fully agree that this represents a compelling avenue for future research, and we thank the reviewer for this valuable input.

Reviewer #3 (Remarks to the Author):

The manuscript "Spatiotemporal profiling of modification-specific proteome secretion uncovers an itaconation-activated tyrosine kinase" by Lu and colleagues introduces PTM-TransitID which is a chemoproteomic technique that uses bioorthogonal probes and advanced mass spectrometry to identify secreted, PTM-bearing proteins. The authors used this technique to identify itaconated proteins and discovered itaconation of tyrosine kinase FYN that enhances its activity.

This study is well executed and offers insights into the role of itaconate in protein modification. However, the current manuscript does not convincingly demonstrate whether itaconate drives the observed effects or if they are predominantly mediated by octyl-itaconate (probes are based on OI rather than itaconate), which possesses distinct biological activities compared to itaconate. Further data and clarifications are necessary to fully validate the findings.

Response: We thank the reviewer for the positive comments.

1. The authors used ITalk (previously characterized in their earlier publication (18)) which is based on octyl-itaconate (OI). This distinction should be clearly stated and discussed, and the title might be adjusted accordingly. Since OI and itaconate are metabolically and functionally distinct (with different electrophilic properties), the authors should carefully clarify these differences and appropriately temper their interpretations and conclusions.

Response: We thank the reviewer for this insightful comment. In response, we have added a discussion on the potential influence of the chemical probes on protein secretion and have emphasized the necessity of orthogonal validation in future work. We have also moderated our conclusions accordingly and adjusted section titles. Regarding the manuscript title, we believe it remains appropriate as it highlights itaconation on FYN, a specific substrate that we have validated using natural itaconate.

2. The experimental model relies on RAW264.7 macrophage-like cells which is a proliferative cell line. To enhance the physiological relevance of key findings, particularly those involving the kinase FYN, some key findings should be repeated in primary macrophages. Some assays were conducted in HEK293T cells (Figure 5) but these key findings were not repeated in RAW264.7 cells.

Response: We thank the reviewer for raising this point. In direct response to the comment, we have now examined the effects of itaconate and 4-OI on FYN phosphorylation at Tyr531 in primary mouse BMDMs. Our results demonstrate that both compounds are potent inhibitors of this phosphorylation event (**Rebuttal Figure 3; Figure 5h in the revised manuscript**). The site-validation experiments were conducted in HEK293T cells because of the intractable difficulty of transfecting Raw264.7 macrophages, a practice consistent with previous studies^{3,4}.

Rebuttal Figure 3. The impact of itaconate and 4-OI on FYN phosphorylation at Tyr531 in BMDMs.

3. RAW264.7 cells are highly responsive to external stimuli which can induce endogenous itaconate production. The authors should clarify whether endogenous itaconate synthesis occurs under their experimental conditions in RAW264.7 cells or whether itaconate is derived from ITalk. The authors might consider employing cells deficient in IRG1 or implementing metabolic tracing approaches.

Response: We thank the reviewer for raising this important point. As suggested, we extracted metabolites from cells following the complete pulse-chase procedure and analyzed them via LC-MS/MS to quantify itaconate levels (**Rebuttal Figure 4; Figure 1b in the revised manuscript**). The results confirm that the labeling process does not interfere with endogenous itaconate metabolism. As a positive control, we confirmed that LPS stimulation significantly increases itaconate concentration, as expected.

Rebuttal Figure 4. Effect of the PBSP workflow on intracellular itaconate levels. a) Schematic of the pulse-chase workflow. Metabolites were extracted from Raw264.7 cells following the complete pulse-chase procedure and analyzed by LC-MS/MS to quantify itaconate levels. For the LPS control, Raw264.7 cells were treated with 100 ng/mL LPS for 14 hours. b) Relative itaconate intensity from three biologically independent experiments. The error bars show mean \pm SD. *** $p < 0.001$ (Student's t-test).

4. Figure 5f depicts effects of exogenous itaconate treatment on FYN. Does endogenously synthesized itaconate, for instance following LPS stimulation, produce comparable effects?

Response: We thank the reviewer for raising this important point. As suggested, we examined IRG1-knockout Raw264.7 cells (**Rebuttal Figure 5a; Figure S5d in the revised manuscript**). We found that LPS treatment did not reduce FYN phosphorylation at Tyr531 in the absence of IRG1 (**Rebuttal Figure 5b; Figure 5j in the revised manuscript**). In comparison, LPS treatment in WT cells significantly inhibits FYN phosphorylation at Tyr531. Together, these data demonstrate that endogenous itaconate is necessary for LPS-mediated FYN activation.

Rebuttal Figure 5. Effect of endogenously synthesized itaconate on FYN Tyr531 phosphorylation. a) Validation of IRG1 knockout in Raw264.7 cells. **b)** FYN Tyr531 phosphorylation in WT and IRG1 KO cells following LPS treatment.

References

1. Huang, Z. et al. Bioorthogonal Photocatalytic Decaging-Enabled Mitochondrial Proteomics. *J. Am. Chem. Soc.* 143(44), 18714–18720. (2021)
2. Runtsch, M. C. et al. Itaconate and itaconate derivatives target JAK1 to suppress alternative activation of macrophages. *Cell Metabolism* 34, 487–501.e488 (2022).
3. Su C, Cheng T, Huang J, Zhang T, Yin H. 4-Octyl itaconate restricts STING activation by blocking its palmitoylation. *Cell Reports.* 42(9):113040.(2023)
4. Wei, Chao. et al. Itaconate protects ferroptotic neurons by alkylating GPx4 post stroke. *Cell death and Differentiation*, 31,983–998.(2024)
5. Qin, W. et al. S-glycosylation-based cysteine profiling reveals regulation of glycolysis by itaconate. *Nature Chemical Biology* 15, 983–991.(2019).
6. Qin, W. et al. Chemoproteomic Profiling of Itaconation by Bioorthogonal Probes in Inflammatory Macrophages. *Journal of the American Chemical Society* 142, 10894–10898.(2020).
7. Branon, T. et al. Efficient proximity labeling in living cells and organisms with TurboID. *Nature Biotechnology.* 36(9):880-887.(2018)

Below are our point-by-point responses to the questions and concerns raised by the reviewers (colored in blue).

Reviewer #1 (Remarks to the Author):

The authors have a good job in addressing the suggestions and concerns from the reviewers and I support its publication

Response: We thank the reviewer for the positive comments.

Reviewer #2 (Remarks to the Author):

I commend the authors for their thorough revisions. My concerns have been fully addressed, and I now support publication of the manuscript in Nature Communications. I believe PBSP has potential for broad application within the field.

Response: We thank the reviewer for the positive comments.

Reviewer #3 (Remarks to the Author):

The reviewers have addressed my concerns. The manuscript now includes data on itaconate and its impact on protein phosphorylation.

The authors also include RAW264.7 IRG1-KO cells. However, I cannot find any additional details on this cell line, including how it was engineered.

Response: We thank the reviewer for the positive comments. The RAW264.7 IRG1-KO cell line was kindly gifted by the Chu Wang Lab at Peking University, which was generated in their previous study¹. We have provided additional information about this cell line in the methods and acknowledgments sections.

References

1. Zihua Liu, Dongyang Liu, Chu Wang, In situ chemoproteomic profiling reveals itaconate inhibits de novo purine biosynthesis in pathogens. Cell Rep, 43(9):114737(2024)